# Analytic Energy-Guided Policy Optimization for Offline Reinforcement Learning

**Jifeng Hu**[1]  **Sili Huang**[2]*  **Zhejian Yang**[1]  **Shengchao Hu**[3]
**Li Shen**[4]  **Hechang Chen**[1]*  **Lichao Sun**[5]
**Yi Chang**[1]*  **Dacheng Tao**[6]
[1]Jilin University  [2]Minzu University of China
[3]Shanghai Jiao Tong University  [4]Shenzhen Campus of Sun Yat-sen University
[5]Lehigh University  [6]Nanyang Technological University
{hujf21, zjyang22}@mails.jlu.edu.cn {chenhc, yichang}@jlu.edu.cn
huangsili@muc.edu.cn mathshenli@gmail.com charles-hu@sjtu.edu.cn
lis221@lehigh.edu dacheng.tao@gmail.com

## Abstract

Conditional decision generation with diffusion models has shown powerful competitiveness in reinforcement learning (RL). Recent studies reveal the relation between energy-function-guidance diffusion models and constrained RL problems. The main challenge lies in estimating the intermediate energy, which is intractable due to the log-expectation formulation during the generation process. To address this issue, we propose the Analytic Energy-guided Policy Optimization (AEPO). Specifically, we first provide a theoretical analysis and the closed-form solution of the intermediate guidance when the diffusion model obeys the conditional Gaussian transformation. Then, we analyze the posterior Gaussian distribution in the log-expectation formulation and obtain the target estimation of the log-expectation under mild assumptions. Finally, we train an intermediate energy neural network to approach the target estimation of log-expectation formulation. We apply our method in 30+ offline RL tasks to demonstrate the effectiveness of our method. Extensive experiments illustrate that our method surpasses numerous representative baselines in D4RL offline reinforcement learning benchmarks.

## 1 Introduction

Controllable generation with diffusion models has shown remarkable success in text-to-image generation [9], photorealistic image synthesization [87], high-resolution video creation [8], and robotics manipulation [10]. A common strategy to realize controllable diffusion models is guided sampling, which can be further classified into two categories: classifier-guided [17, 48] and classifier-free-guided [61, 1] diffusion-based methods.

Usually, classifier guidance and classifier-free guidance need paired data, i.e., samples and the corresponding conditioning variables, to train a controllable diffusion model [85, 24, 74]. However, it is difficult to describe the conditioning variables for each transition in RL. We can only evaluate the value for the transitions with a scalar and continuous function, which is generally called the action value function (Q function) [64, 48]. Recently, energy-function-guided diffusion models provide an effective way to realize elaborate manipulation in RL. Previous studies also reveal the relation of equivalence between guided sampling energy-function-guided diffusion models and constrained RL

---

*Corresponding authors: Hechang Chen, Sili Huang, and Yi Chang.
code: https://github.com/JF-Hu/Analytic-Energy-guided-Policy-Optimization

39th Conference on Neural Information Processing Systems (NeurIPS 2025).

problems [64, 11]. Considering the following formula of energy-function-guided diffusion models

$$\min_{p} \ \mathbb{E}_{x \sim p(x)} \mathcal{E}(x),$$
$$s.t. \ D_{KL}(p(x)\|q(x)), \tag{1}$$

where $x \in \mathbb{R}^d$, $\mathcal{E}(x) : \mathbb{R}^d \to \mathbb{R}$ is the energy function, $q(x)$ and $p(x)$ denote the unguided and guided data distribution, and $D_{KL}(\cdot)$ represents the KL divergence. The optimal solution to the above problem is

$$p(x) \propto q(x)e^{-\beta \mathcal{E}(x)}, \tag{2}$$

where inverse temperature $\beta$ is the Lagrangian multiplier, which controls the energy strength. Equation (2) shows that the guided data distribution lies in the intersection region of unguided distribution $q(x)$ and energy distribution $e^{-\beta \mathcal{E}(x)}$. Obversely, the guidance comes from the energy function $\mathcal{E}(x)$. The constrained RL problem

$$\max_{\pi} \ \mathbb{E}_{s \sim D^{\mu}} \left[ \mathbb{E}_{a \sim \pi(\cdot|s)} Q(s,a) \right],$$
$$s.t. \ D_{KL}(\pi(\cdot|s)\|\mu(\cdot|s)) < \varepsilon, \tag{3}$$

has the similar formula with Equation (1), where $\pi$ and $\mu$ are the learned and behavior policies. Thus,

$$\pi^*(a|s) \propto \mu(a|s)e^{\beta Q(s,a)}. \tag{4}$$

The optimal policy $\pi^*(a|s)$ under state $s$ can be regarded as the guided distribution $p(x)$, and actions $a$ can be generated with guided sampling. See Section 6.2 for more discussion and Appendix C for detailed derivation.

Although sampling from $p(x)$ is intractable due to the normalization term, it can be bypassed in score-based diffusion models, where the score function $\nabla_x \log p(x)$ can be calculated with intermediate energy $\mathcal{E}_t(x_t)$ (Refer to Equation (8) for details.) [61, 28]. However, the intermediate energy introduced in classifier-guided and classifier-free-guided methods has proved to be inexact theoretically and practically [64, 90]. Besides, for classifier-guided methods, the intermediate guidance can be influenced by the fictitious state-action pairs generated during the generation process. For classifier-free-guided methods, the intermediate guidance is manually predefined during guided sampling. Though recent studies [64] propose the exact expression of $\mathcal{E}_t(x_t)$, they do not investigate the theoretical solution of the intermediate energy.

In this paper, we theoretically analyze the intermediate energy $\mathcal{E}_t(x_t)$ (intermediate guidance is $\nabla_{x_t} \mathcal{E}_t(x_t)$) with a log-expectation formulation and derive the solution of intermediate energy by addressing the posterior integral. Based on the theoretical results, we propose a new diffusion-based method called Analytic Energy-guided Policy Optimization (AEPO). In contrast with previous studies, we leverage the characteristic of Gaussian distribution and find a solution for the log-expectation formulation. Specifically, we first convert the implicit dependence on the action in the exponential term to explicit dependence by applying Taylor expansion. After that, we investigate the posterior distribution formulation of the expectation term and simplify the intractable log-expectation formulation with the moment-generating function of the Gaussian distribution. Then, we propose to train a general Q function and an intermediate energy function to approach the simplified log-expectation formulation given a batch of data. Finally, during the inference, we can use the intermediate guidance to generate guided information to high-return action distributions.

To verify the effectiveness of our method, we apply our method in offline RL benchmarks D4RL [21], where we select different tasks with various difficulties. We compare our method with dozens of baselines, which contain many types of methods, such as classifier-guided and classifier-free-guided diffusion models, behavior cloning, and transformer-based models. Through extensive experiments, we demonstrate that our method surpasses state-of-the-art algorithms in most environments.

## 2 Related Work

### 2.1 Offline RL

Offline RL aims to learn a policy entirely from previously collected datasets, thus avoiding expensive and risky interactions with the environment, such as autonomous driving [56, 35, 53, 30, 60, 44, 45].

However, in practice, we may face the distribution shift issue, which means that the learned policy and the behavior policy are different, and the overestimation of out-of-distribution (OOD) actions will lead to a severe performance drop [55, 27, 22, 32, 71, 42, 41, 46]. In order to solve these issues, previous studies can be roughly classified into several research lines. Policy regularization methods [7, 66, 88, 22, 54] focus on applying constraints to the learned policy to prevent it from deviating far from the behavior policy. Critic penalty methods [55, 53, 65, 68, 43] propose to train a conservative value function that assigns low expected return on unseen samples and high return on dataset samples, resulting in efficient OOD actions overestimation. Uncertainty quantification methods [89, 2, 4, 83] introduce uncertainty estimation to identify whether an action is OOD, thus enhancing the robustness on OOD actions. Recently, generative models [26, 57, 81, 39, 37, 38, 15], such as diffusion models, are proposed to augment the offline datasets with synthetic experiences or train a world model for planning.

## 2.2 Generative Policy Optimization

Recent advancements [48] in diffusion RL methods have shown the diffusion models' powerful expression in modeling multimodal policies [11, 82, 34], learning heterogeneous behaviors [1, 25, 58], and generating fine-grained control [40, 16, 33]. Diffusion-based RL algorithms cursorily contain two types of guided sampling to realize policy optimization [12]. Guidance-based policy optimization [48, 64, 91] uses the value function to instruct the action generation process of diffusion models. During the intermediate diffusion steps, the generated action vectors will incline to the region with high values according to the intermediate guidance. Selected-based policy optimization [11, 25, 67] first generates a batch of candidate actions from diffusion behavior policy. Then, it constructs a critic-weighted empirical action distribution and resamples the action from it for evaluation.

## 3 Preliminary

### 3.1 Diffusion Probabilistic Models

Diffusion Probabilistic Models (DPMs) [29, 17, 74] are proposed to construct the transformation between complex data distribution $x_0$ and easy sampling distribution $x_T$ (e.g., Gaussian distribution). By defining the forward transformation from data distribution $x_0$ to simple distribution $x_T$ in $T$ time range according to the following stochastic differential equation $dx = f(x, t)dt + g(t)dw$, where $f(\cdot, t) : \mathbb{R}^d \to \mathbb{R}^d$ is the drift coefficient, $g(\cdot)$ is the diffusion coefficient, and w is the standard Wiener process, we can obtain a reverse transformation from the simple distribution $x_T$ to data distribution $x_0$, as shown in $dx = \left[ f(x, t) - g(t)^2 \nabla_x \log q_t(x) \right] dt + g(t)d\bar{w}$, where $\bar{w}$ is the standard Wiener process, $t \in [0, T]$ is the diffusion time, $t = 0$ means data without perturbation, $t = T$ means prior sample-friendly distribution, $dt$ indicates infinitesimal negative timestep, $q_t(x)$ is the marginal distribution of $x_t$, and $\nabla_x \log q_t(x)$ is the score function. Usually, we adopt the Gaussian transition distribution [62, 63], which means

$$q_{t|0}(x_t|x_0) = \mathcal{N}(x_t; \alpha_t x_0, \sigma_t^2 \boldsymbol{I}), \tag{5}$$

where $\alpha_t > 0$ is a decrease function, $\sigma_t > 0$ is an increase function, $q_{T|0}(x_T|x_0) \approx \mathcal{N}(x_T; 0, \tilde{\sigma}^2 \boldsymbol{I})$ for certain $\tilde{\sigma}$. Obviously, $q_{T|0}(x_T|x_0)$ will be independent of $x_0$, i.e., $q_{T|0}(x_T|x_0) \approx q_T(x_T)$. Based on the above results, once we obtain $\nabla_x \log q_t(x)$, we can perform generation from simple distribution [5, 86]. By constructing the score function $s(x_t)$, previous studies [78, 77] show that the following objective is equivalent: $\mathbb{E}_{q(x_t)} \left[ ||s(x_t) - \nabla_{x_t} \log q_t(x_t)||_2^2 \right] \Rightarrow \mathbb{E}_{q(x_0)q_{t|0}(x_t|x_0)} \left[ ||s(x_t) - \nabla_{x_t} \log q_{t|0}(x_t|x_0)||_2^2 \right]$ and the reverse transformation can be an alternative formula, i.e., probability flow ordinary differential equation (ODE)

$$\frac{dx_t}{dt} = f(t)x_t - \frac{1}{2}g(t)^2 \nabla_{x_t} \log p_t(x_t), \tag{6}$$

where $f(x, t) = f(t)x_t$, $f(t) = \frac{d \log \alpha_t}{dt}$, and $g(t)^2 = \frac{d \sigma_t^2}{dt} - 2\frac{d \log \alpha_t}{dt}\sigma_t^2$, because Equation (6) holds the marginal distribution unaltered [52]. Considering that $x_t = \alpha_t x_0 + \sigma_t \epsilon$, where $\epsilon \sim \mathcal{N}(0, \boldsymbol{I})$, there exist $s(x_t) = \nabla_{x_t} \log p_t(x_t) \approx -\frac{\epsilon}{\sigma_t}$. So, we can introduce a neural network $\epsilon_\theta(x_t, t)$, and

$$\mathcal{L}_{diff} = \mathbb{E}_{x_0 \sim q(x_0), t \sim U(0,T)} \left[ ||\epsilon_\theta(x_t, t) - \epsilon||_2^2 \right], \tag{7}$$

is the diffusion loss, where $U(\cdot)$ is uniform distribution. $x_0 \sim q(x_0)$ means sampling data from the offline datasets. After training, we can obtain the score function value through $\epsilon_\theta$, i.e., $\nabla_{x_t} \log p_t(x_t) \approx -\frac{\epsilon_\theta}{\sigma_t}$. Then, we can use Equation (6) for the generation process [62].

## 3.2 Guided Sampling

In RL, guidance plays an important role in generating plausible decisions because the dataset quality is usually mixed, and naive modeling of the conditional action distribution under states will lead to suboptimal performance. Classifier-guided methods [17, 82, 48, 50] define binary random optimality variables $\mathcal{O}$, where $\mathcal{O} = 1$ means optimal outputs, and $\mathcal{O} = 0$ means suboptimal outputs. For each generation step $t$, $p(x_{t-1}|x_t, \mathcal{O}) \propto q(x_{t-1}|x_t)p(\mathcal{O}|x_t)$, the guidance serves as a gradient on the mean value modification $p(x_{t-1}|x_t, \mathcal{O}) = \mathcal{N}(x_t; \mu_t + \Sigma_t \cdot \nabla \log p(\mathcal{O}|x_t), \Sigma_t)$, where $\mu_t$ and $\Sigma_t$ is the predicted mean and variances of $q(x_{t-1}|x_t)$. During inference, the fictitious intermediate outputs $x_t$ that do not exist in the training dataset will lead to suboptimal intermediate guidance. Different from classifier-guided methods, the classifier-free-guided methods implicitly build the joint distribution between the data and condition variables $\mathcal{C}$ in the training phase [1, 11, 61, 12]. Therefore, we can use the desired condition as guidance to perform guided sampling. The classifier-free-guided training loss is $\mathbb{E}_{x_0 \sim q(x_0), t \sim U(0,T), b \sim \mathcal{B}(\lambda)} \left[||\epsilon_\theta(x_t, b * \mathcal{C}, t) - \epsilon||_2^2\right]$, where $\mathcal{B}$ is binomial distribution, $\lambda$ is the parameter of $\mathcal{B}$. During inference, the guidance is also incorporated in the mixed prediction of conditional and unconditional noise, i.e., $\hat{\epsilon} = \epsilon_\theta(x_t, t, \emptyset) + \omega(\epsilon_\theta(x_t, t, \mathcal{C}) - \epsilon_\theta(x_t, t, \emptyset))$, $\omega$ controls the guidance strength, $\emptyset$ means $b = 0$. Due to the condition variables being needed before generation, we need to manually assign the value before inference. The predefined $\mathcal{C}$ will also restrict the model's performance if insufficient prior information on environments arises.

## 3.3 Diffusion Offline RL

Typical RL [69] is formulated by the Markov Decision Process (MDP) that is defined as the tuple $\mathcal{M} = \langle \mathcal{S}, \mathcal{A}, \mathcal{P}, r, \gamma \rangle$, where $\mathcal{S}$ and $\mathcal{A}$ denote the state and action space, respectively, $\mathcal{P}(s'|s, a)$ is the Markovian transition probability, $r(s, a)$ is the reward function, and $\gamma \in [0, 1)$ is the discount factor. The goal is to find a policy $\pi$ that can maximize the discounted return $\mathbb{E}_\pi[\sum_{k=0}^\infty \gamma^k r(s_k, a_k)]$, where $k$ represents the RL time step which is different from diffusion step $t$ [76]. In offline RL [55, 53], only a static dataset $D_\mu$ collected with behavior policy is available for training. Extracting optimal policy from offline datasets is formulated as a constrained RL problem [72, 84] as shown in Equation (3), which can be converted to find an optimal policy $\pi^*$ that maximizes $\pi^* = \max_\pi \mathbb{E}_{s \sim D_\mu} \mathbb{E}_{a \sim \pi(\cdot|s)} \left[Q(s, a) - \frac{1}{\beta} D_{KL}(\pi(\cdot|s)||\mu(\cdot|s))\right]$. Diffusion offline RL usually adopts diffusion models to imitate the behavior policy $\mu$ [1]. Naively modeling the action distribution will only lead to suboptimal policy. Thus, the well-trained value functions will be used to extract optimal policy, such as action value gradient guidance [48] and empirical action distribution reconstruction [25].

# 4 Method

In order to generate samples from the desired distribution $p(x)$ with the reverse transformation (generation process) Equation (6), we should know the score function $\nabla_x \log p(x)$. If the score function of the desired distribution $p(x)$ has a relation with the score function of $q(x)$

$$\nabla_x \log p(x) = \nabla_x \log q(x) + \nabla_x - \beta \mathcal{E}(x), \tag{8}$$

for any data $x$, we can obtain the score function of $p(x)$ by compounding the score function of $q(x)$ and the gradient of $\mathcal{E}(x)$. However, it only exists $p_0(x_0) \propto q_0(x_0)e^{-\beta\mathcal{E}(x_0)}$ for the samples in the dataset rather than the marginal distribution of $p_t(x_t)$ and $q_t(x_t)$. Previous guided sampling methods usually adopt MSE or diffusion posterior sampling (DPS) as the objective of training the intermediate energy, which cannot satisfy the relation $p_t(x_t) \propto q_t(x_t)e^{-\mathcal{E}_t(x_t)}$, thus leading to an inexact intermediate guidance. We summarize the results in Theorem 4.1.

**Theorem 4.1** (Inexact and Exact Intermediate Energy). *Suppose $p_0(x_0)$ and $q_0(x_0)$ has the relation of Equation* (2). *$p_{t|0}(x_t|x_0)$ and $q_{t|0}(x_t|x_0)$ are defined by $p_{t|0}(x_t|x_0) := q_{t|0}(x_t|x_0) = \mathcal{N}(x_t; \alpha_t x_0, \sigma_t^2 \boldsymbol{I})$ for all $t \in (0, T]$. According to the Law of Total Probability, the marginal*

distribution $p_t(x_t)$ and $q_t(x_t)$ are given by $p_t(x_t) = \int p_{t|0}(x_t|x_0)p_0(x_0)dx_0$ and $q_t(x_t) = \int q_{t|0}(x_t|x_0)q_0(x_0)dx_0$. Previous studies [64, 67, 14] define the inexact intermediate energy as

$$\mathcal{E}_t^{MSE}(x_t) = \mathbb{E}_{q_{0|t}(x_0|x_t)}[\mathcal{E}(x_0)], t > 0; \quad \mathcal{E}_t^{DPS}(x_t) = \mathcal{E}(\mathbb{E}_{q_{0|t}(x_0|x_t)}[x_0]), t > 0. \tag{9}$$

*The exact intermediate energy is defined by*

$$\mathcal{E}_t(x_t) = -\log \mathbb{E}_{q_{0|t}(x_0|x_t)}[e^{-\beta\mathcal{E}(x_0)}], t > 0. \tag{10}$$

*It can be proved (See Appendix D for details.) that $p_t(x_t) \propto q_t(x_t)e^{-\mathcal{E}_t(x_t)}$ exists under Equation (10) rather than Equation (9). So, the intermediate guidance (Equation (8)) is inexact for previous classifier-guided and classifier-free-guided methods.*

Similarly, in RL, as shown in Equation (4), the desired distribution is $\pi(a|s)$. The intermediate energy $\mathcal{E}_t(s, a_t)$ and the score function $\nabla_{a_t} \log \pi_t(a_t|s)$ with intermediate guidance $\nabla_{a_t}\mathcal{E}_t(s, a_t)$ in RL are defined as follows

$$\mathcal{E}_t(s, a_t) = \begin{cases} \beta Q(s, a_0), & t = 0 \\ \log \mathbb{E}_{\mu_{0|t}(a_0|a_t, s)}[e^{\beta Q(s,a_0)}], & t > 0 \end{cases} \tag{11}$$

$$\nabla_{a_t} \log \pi_t(a_t|s) = \nabla_{a_t} \log \mu_t(a_t|s) + \nabla_{a_t}\mathcal{E}_t(s, a_t), \tag{12}$$

where we slightly abuse the input of $\mathcal{E}$ because the value action should depend on the corresponding state in RL. Refer to Appendix E for the detailed derivation.

## 4.1 Intermediate Energy

Diffusion loss (Equation (7)) provides the way to obtain the score function of $\mu_t(a_t|s)$,

$$\nabla_{a_t} \log \mu_t(a_t|s) = -\frac{\epsilon_\theta(s, a_t, t)}{\sigma_t}. \tag{13}$$

Obviously, the most challenging issue that we need to address is $\log \mathbb{E}_{\mu_{0|t}(a_0|a_t,s)}[e^{\beta Q(s,a_0)}]$ because of the intractable log-expectation formulation. Following previous research, we use a Gaussian distribution as the estimate for the posterior distribution $\mu_{0|t}(a_0|a_t, s)$, where the mean and covariance are denoted as $\mu_{0|t} = \mathcal{N}(\tilde{\mu}_{0|t}, \tilde{\Sigma}_{0|t})$. We will introduce how to approximate $\tilde{\mu}_{0|t}$ and $\tilde{\Sigma}_{0|t}$ below. We first focus on converting the implicit dependence on the action in the exponential term to explicit dependence by applying Taylor expansion. Expand $Q(s, a_0)$ at $a_0 = \bar{a}$ with Taylor expansion, where $\bar{a}$ is a constant vector. Then, we have

$$Q(s, a_0) \approx Q(s, a_0)|_{a_0=\bar{a}} + \frac{\partial Q(s, a_0)}{\partial a_0}^\top |_{a_0=\bar{a}} * (a_0 - \bar{a}). \tag{14}$$

Replacing Q function in $\log \mathbb{E}_{a_0\sim\mu(a_0|a_t,s)}e^{\beta Q(s,a_0)}$ with Equation (14), we will derive the following

$$\log \mathbb{E}_{a_0\sim\mu(a_0|a_t,s)}e^{\beta Q(s,a_0)} \approx \beta Q(s, \bar{a}) - \beta Q'(s, \bar{a})^\top \bar{a} + \log \left\{ \mathbb{E}_{a_0\sim\mu(a_0|a_t,s)}[e^{\beta Q'(s,\bar{a})^\top a_0}] \right\}, \tag{15}$$

where $Q' = \frac{\partial Q}{\partial a}$. More discussion about the approximation of intermediate energy can be found in Appendix F and G. Note that the above derivation makes the complex dependence between $Q(s, a_0)$ and $a_0$ easier. In other words, we transfer $a_0$ from the implicit dependence on $Q(s, a_0)$ to explicit dependence on $Q'(s, \bar{a})^\top a_0$. Refer to Appendix H for more details. As for the only unknown item $\mathbb{E}_{a_0\sim\mu(a_0|a_t,s)}[e^{\beta Q'(s,\bar{a})^\top a_0}]$, the exact result can be derivated from moment generating function, which indicates that $\mathbb{E}_{x\sim\mathcal{N}(v,\Sigma)}[e^{a^\top x}] = e^{a^\top v + \frac{1}{2}a^\top \Sigma a}$. By simplifying Equation (15), we have

$$\log \left\{ \mathbb{E}_{a_0\sim\mu(a_0|a_t,s)}[e^{\beta Q'(s,\bar{a})^\top a_0}] \right\} = \beta Q'(s, \bar{a})^\top \tilde{\mu}_{0|t} + \frac{1}{2}\beta^2 Q'(s, \bar{a})^\top \tilde{\Sigma}_{0|t} Q'(s, \bar{a}).$$

Finally, the intermediate energy $\mathcal{E}_t(s, a_t)$ with log-expectation formulation can be approximated by

$$\log \mathbb{E}_{a_0\sim\mu(a_0|a_t,s)}e^{\beta Q(s,a_0)} \approx \beta Q(s, \bar{a}) + \beta Q'(s, \bar{a})^\top (\tilde{\mu}_{0|t} - \bar{a}) + \frac{1}{2}\beta^2 Q'(s, \bar{a})^\top \tilde{\Sigma}_{0|t} Q'(s, \bar{a}). \tag{16}$$

From Equation (16), we can see that the intermediate energy can be approximated with $\bar{a}$, $\tilde{\mu}_{0|t}$, and $\tilde{\Sigma}_{0|t}$. In the next section, we will introduce how to approximate the parameters of the posterior distribution $\mu(a_0|a_t, s)$. The training algorithm is shown in Algorithm 1 of Appendix A.

## 4.2 Posterior Approximation

In this paper, we provide several methods to approximate the distribution of $\mu(a_0|a_t, s)$. We use "Posterior i" to differentiate different posterior approximation methods in the following contents.

**Posterior 1.** Inspired by previous studies [6], we can use the trained diffusion model $\epsilon_\theta$ to obtain the mean vector $\tilde{\mu}_{0|t}$ of the distribution $\mu_{0|t}(a_0|a_t, s) = \mathcal{N}(a_0; \tilde{\mu}_{0|t}, \tilde{\Sigma}_{0|t})$, where the mean vector

$$\tilde{\mu}_{0|t} = \frac{1}{\alpha_t}(a_t - \sigma_t \epsilon_\theta(s, a_t, t)) \tag{17}$$

according to Equation (5). As for covariance matrix $\tilde{\Sigma}_{0|t}$, following the definition of covariance

$$\tilde{\Sigma}_{0|t}(a_t) = \frac{1}{\alpha_t^2} \mathbb{E}_{\mu_{0|t}(a_0|a_t, s)} \left[ (a_t - \alpha_t a_0)(a_t - \alpha_t a_0)^\top \right] - \frac{\sigma_t^2}{\alpha_t^2} \epsilon_\theta \epsilon_\theta^\top, \tag{18}$$

where in the second equation we use $\mathbb{E}_{\mu_{0|t}(a_0|a_t)}[(a_0 - \frac{1}{\alpha_t}a_t) * \frac{\sigma_t}{\alpha_t}\epsilon_\theta] = (\mathbb{E}_{\mu_{0|t}(a_0|a_t, s)}[a_0] - \frac{1}{\alpha_t}a_t) * \frac{\sigma_t}{\alpha_t}\epsilon_\theta = (\tilde{\mu}_{0|t} - \frac{1}{\alpha_t}a_t) * \frac{\sigma_t}{\alpha_t}\epsilon_\theta = -\frac{\sigma_t^2}{\alpha_t^2}\epsilon_\theta \epsilon_\theta^\top$, $\epsilon_\theta = \epsilon_\theta(s, a_t, t)$, and $\tilde{\mu}_{0|t} = \mathbb{E}_{\mu_{0|t}(a_0|a_t, s)}[a_0]$. Besides, we also notice that $\mu(a_t) \sim \mathcal{N}(a_t; \alpha_t a_0, \sigma_t^2 I)$, thus we have

$$\mathbb{E}_{\mu_t(a_t|s)} \mathbb{E}_{\mu_{0|t}(a_0|a_t, s)} \left[ (a_t - \alpha_t a_0)(a_t - \alpha_t a_0)^\top \right] = \sigma_t^2 I. \tag{19}$$

To simplify the problem of calculating the exact covariance matrix, we adopt two strategies. 1) The isotropic Gaussian assumption: Computing an exact covariance matrix is expensive (if the matrix size is $N$, it requires $N^2$ computation times.). Under the isotropic Gaussian assumption, we only need N computation times, which significantly reduces the computational cost. 2) The marginalization over $a_t$: Since $a_t$ is sampled from a distribution, data points far from the mean can lead to instability during training. Therefore, we marginalize over $a_t$ to use the average effect to replace the specific effect of each individual $a_t$ on $a_0$. Based on the above two strategies, we have

$$\tilde{\Sigma}_{0|t} = \mathbb{E}_{\mu_t(a_t|s)} \tilde{\Sigma}_{0|t}(a_t) = \frac{\sigma_t^2}{\alpha_t^2}[I - \mathbb{E}_{\mu_t(a_t|s)}[\epsilon_\theta \epsilon_\theta^\top]]. \tag{20}$$

For simplicity, we usually consider isotropic Gaussian distribution, where the covariance $\tilde{\Sigma}_{0|t}$ satisfies $\tilde{\Sigma}_{0|t} = \tilde{\sigma}_{0|t}^2 * I$, the covariance can be simplified as

$$\tilde{\sigma}_{0|t}^2 = \frac{\sigma_t^2}{\alpha_t^2} \left[ 1 - \frac{1}{d}\mathbb{E}_{\mu_t(a_t|s)} \left[ ||\epsilon_\theta(a_t, t)||_2^2 \right] \right]. \tag{21}$$

In the experiments, we adopt this posterior as the default setting to conduct all experiments. Refer to Appendix I.1 for the detailed derivation.

**Posterior 2.** As Equation (17) shows, we still adopt the same mean of the distribution $\mu_{0|t}(a_0|a_t, s)$. We assume that the $q_{0|t}(a_0|a_t)$ obey the formulation $\mu_{0|t}(a_0|a_t, s) = \mathcal{N}(a_0; \tilde{\mu}_{0|t}, \tilde{\Sigma}_{0|t})$. Again considering Equation (18), we reformulate [79] it as

$$\tilde{\Sigma}_{0|t}(a_t) = \mathbb{E}_{\mu_{0|t}(a_0|a_t, s)} \left[ (a_0 - u_0)(a_0 - u_0)^\top \right] - (\tilde{\mu}_{0|t} - u_0)(\tilde{\mu}_{0|t} - u_0)^\top, \tag{22}$$

where $u_0$ is any constant vector, which has the same dimension with $\tilde{\mu}_{0|t}$. When we consider isotropic Gaussian distribution and remove the inference of $a_t$, the $\tilde{\Sigma}_{0|t}$ is simplified as $\tilde{\Sigma}_{0|t} = \tilde{\sigma}_{0|t}^2 I$ and

$$\tilde{\sigma}_{0|t}^2 = Var(a_0) - \frac{1}{d}\mathbb{E}_{\mu_t(a_t|s)}[||\tilde{\mu}_{0|t} - u_0||_2^2], \tag{23}$$

where $u_0 = \mathbb{E}_{a_0 \sim \mu(a_0)}[a_0]$ and we can use the mean value of the whole action vectors of the dataset as an unbiased estimation. $Var(a_0)$ can be approximated from a batch of data or from the entire dataset. We can sample a batch of data $a_t$ with different $t$ to calculate an approximation solution of the second term $\mathbb{E}_{\mu_t(a_t|s)}[||\tilde{\mu}_{0|t} - u_0||_2^2]$. In Appendix I.2, we provide the detailed derivation. From the above theory, we can solve the Gaussian distribution $\mu_{0|t}(a_0|a_t, s)$'s parameters $(\tilde{\mu}_{0|t}, \tilde{\Sigma}_{0|t})$. According to Equation (16), we have simplified intermediate energy:

$$\mathcal{E}_t(s, a_t) \approx \beta Q(s, \bar{a}) + \beta Q'(s, \bar{a})^\top (\tilde{\mu}_{0|t} - \bar{a}) + \frac{1}{2}\beta^2 \tilde{\sigma}_{0|t}^2 * ||Q'(s, \bar{a})||_2^2, \tag{24}$$

and the intermediate energy training loss $\mathcal{L}_{IE}$ is

$$\mathcal{L}_{IE} = \mathbb{E} \left[ ||\mathcal{E}_\Theta(s, a_t, t) - \mathcal{E}_t(s, a_t)||_2^2 \right], \tag{25}$$

where $\Theta$ is the parameter, the mean and covariance are given by Equation (17), (21), and (23).

Table 1: Offline RL algorithms comparison on D4RL Gym-MuJoCo tasks, where we use `red color` and `blue color` to show diffusion-based and non-diffusion-based baselines. Our method is shown with `yellow color`.

| Dataset | Med-Expert | | | Medium | | | Med-Replay | | | mean score | total score |
|---|---|---|---|---|---|---|---|---|---|---|---|
| Env | HalfCheetah | Hopper | Walker2d | HalfCheetah | Hopper | Walker2d | HalfCheetah | Hopper | Walker2d | | |
| AWAC | 42.8 | 55.8 | 74.5 | 43.5 | 57.0 | 72.4 | 40.5 | 37.2 | 27.0 | 50.1 | 450.7 |
| BC | 55.2 | 52.5 | 107.5 | 42.6 | 52.9 | 75.3 | 36.6 | 18.1 | 26.0 | 51.9 | 466.7 |
| MOPO | 63.3 | 23.7 | 44.6 | 42.3 | 28.0 | 17.8 | 53.1 | 67.5 | 39.0 | 42.1 | 379.3 |
| MBOP | 105.9 | 55.1 | 70.2 | 44.6 | 48.8 | 41.0 | 42.3 | 12.4 | 9.7 | 47.8 | 430.0 |
| MOReL | 53.3 | 108.7 | 95.6 | 42.1 | 95.4 | 77.8 | 40.2 | 93.6 | 49.8 | 72.9 | 656.5 |
| TAP | 91.8 | 105.5 | 107.4 | 45.0 | 63.4 | 64.9 | 40.8 | 87.3 | 66.8 | 74.8 | 672.9 |
| BEAR | 51.7 | 4.0 | 26.0 | 38.6 | 47.6 | 33.2 | 36.2 | 10.8 | 25.3 | 30.4 | 273.4 |
| BCQ | 64.7 | 100.9 | 57.5 | 40.7 | 54.5 | 53.1 | 38.2 | 33.1 | 15.0 | 50.9 | 457.7 |
| CQL | 62.4 | 98.7 | 111.0 | 44.4 | 58.0 | 79.2 | 46.2 | 48.6 | 26.7 | 63.9 | 575.2 |
| TD3+BC | 90.7 | 98.0 | 110.1 | 48.3 | 59.3 | 83.7 | 44.6 | 60.9 | 81.8 | 75.3 | 677.4 |
| IQL | 86.7 | 91.5 | 109.6 | 47.4 | 66.3 | 78.3 | 44.2 | 94.7 | 73.9 | 77.0 | 692.6 |
| PBRL | 92.3 | 110.8 | 110.1 | 57.9 | 75.3 | 89.6 | 45.1 | 100.6 | 77.7 | 84.4 | 759.4 |
| DT | 90.7 | 98.0 | 110.1 | 42.6 | 67.6 | 74.0 | 36.6 | 82.7 | 66.6 | 74.3 | 668.9 |
| TT | 95.0 | 110.0 | 101.9 | 46.9 | 61.1 | 79.0 | 41.9 | 91.5 | 82.6 | 78.9 | 709.9 |
| BooT | 94.0 | 102.3 | 110.4 | 50.6 | 70.2 | 82.9 | 46.5 | 92.9 | 87.6 | 81.9 | 737.4 |
| SfBC | 92.6 | 108.6 | 109.8 | 45.9 | 57.1 | 77.9 | 37.1 | 86.2 | 65.1 | 75.6 | 680.4 |
| D-QL@1 | 94.8 | 100.6 | 108.9 | 47.8 | 64.1 | 82.0 | 44.0 | 63.1 | 75.4 | 75.6 | 680.7 |
| Diffuser | 88.9 | 103.3 | 106.9 | 42.8 | 74.3 | 79.6 | 37.7 | 93.6 | 70.6 | 77.5 | 697.7 |
| DD | 90.6 | 111.8 | 108.8 | 49.1 | 79.3 | 82.5 | 39.3 | 100.0 | 75.0 | 81.8 | 736.4 |
| IDQL | 95.9 | 108.6 | 112.7 | 51.0 | 65.4 | 82.5 | 45.9 | 92.1 | 85.1 | 82.1 | 739.2 |
| HDMI | 92.1 | 113.5 | 107.9 | 48.0 | 76.4 | 79.9 | 44.9 | 99.6 | 80.7 | 82.6 | 743.0 |
| AdaptDiffuser | 89.6 | 111.6 | 108.2 | 44.2 | 96.6 | 84.4 | 38.3 | 92.2 | 84.7 | 83.3 | 749.8 |
| DiffuserLite | 87.8 | 110.7 | 106.5 | 47.6 | 99.1 | 85.9 | 41.4 | 95.9 | 84.3 | 84.4 | 759.2 |
| HD-DA | 92.5 | 115.3 | 107.1 | 46.7 | 99.3 | 84.0 | 38.1 | 94.7 | 84.1 | 84.6 | 761.8 |
| Consistency-AC | 84.3 | 100.4 | 110.4 | 69.1 | 80.7 | 83.1 | 58.7 | 99.7 | 79.5 | 85.1 | 765.9 |
| TCD | 92.7 | 112.6 | 111.3 | 47.2 | 99.4 | 82.1 | 40.6 | 97.2 | 88.0 | 85.7 | 771.0 |
| D-QL | 96.1 | 110.7 | 109.7 | 50.6 | 82.4 | 85.1 | 47.5 | 100.7 | 94.3 | 86.3 | 777.1 |
| QGPO | 93.5 | 108.0 | 110.7 | 54.1 | 98.0 | 86.0 | 47.6 | 96.9 | 84.4 | 86.6 | 779.2 |
| AEPO | 94.4±0.9 | 111.5±1.1 | 109.3±0.5 | 49.6±1.1 | 100.2±0.5 | 86.2±1.1 | 43.7±1.3 | 101.0±0.9 | 90.8±1.5 | 87.4 | 786.7 |

## 4.3 Q Function Training

Due to the fact that $\mathcal{E}_t(s, a_t)$ rely on $Q(s, a)$, we first incorporate an action value function $Q_\psi(s, a)$ with parameters $\psi$ to approximate the action values [36]. As for the training of the $Q_\psi(s, a)$, we leverage the expectile regression loss to train the Q function and V function:

$$
\begin{aligned}
\mathcal{L}_V &= \mathbb{E}_{(s,a)\sim D_\mu}\left[L_2^\tau(V_\phi(s) - Q_{\bar\psi}(s, a))\right], \\
\mathcal{L}_Q &= \mathbb{E}_{(s,a,s')\sim D_\mu}\left[||r(s, a) + \gamma V_\phi(s') - Q_\psi(s, a)||_2^2\right], \\
L_2^\tau(y) &= |\tau - 1(y < 0)|y^2,
\end{aligned}
\tag{26}
$$

where $\phi$ is the parameters of value function $V$, $\bar\psi$ is the parameters of target Q, $\tau$ controls the weights of different $y$. There are also some other methods suitable for learning the Q function from offline datasets, such as In-support Q-learning [64] and conservative Q-learning [55]. However, they use either fake actions that are generated from generative models or over-underestimate values for out-of-dataset actions, which influence the learned Q values and the calculation of the intermediate energy. We defer more discussion of the training of the Q function on offline datasets in Appendix J.

## 4.4 Guidance Rescaling

As shown in Equation (12), the magnitude and direction of $\nabla_{a_t} \log \pi_t(a_t|s)$ will be easily affected by the gradient of the intermediate energy. Previous studies usually need extra hyperparameter $w$ to adjust the guidance degree to find better performance, i.e., $\nabla_{a_t} \log \pi_t(a_t|s) = \nabla_{a_t} \log \mu_t(a_t|s) + w\nabla_{a_t}\mathcal{E}_t(s, a_t)$. However, it poses issues to the performance stability of different $w$ during inference. Inspired by the experimental phenomenon that when the guidance scale is zero, the inference performance is more stable than that when the guidance scale is non-zero. We propose to re-normalize the magnitude of $\nabla_{a_t} \log \pi_t(a_t|s)$ by $\nabla_{a_t} \log \pi_t(a_t|s) = \frac{\nabla_{a_t} \log \pi_t(a_t|s)}{||\nabla_{a_t} \log \pi_t(a_t|s)||} * ||\nabla_{a_t} \log \mu_t(a_t|s)||$.

## 5 Experiments

In the following sections, we report the details of environmental settings, evaluation metrics, and comparison results.

Table 2: D4RL Pointmaze (maze2d) and Locomotion (antmaze) comparison. We select 12 subtasks for evaluation, including different difficulties and reward settings. We use red color , blue color , and yellow color to show diffusion-based baselines, non-diffusion-based baselines, and our method.

| Environment | maze2d-umaze | | maze2d-medium | | maze2d-large | | mean sparse score | mean dense score |
|---|---|---|---|---|---|---|---|---|
| Environment type | sparse | dense | sparse | dense | sparse | dense | | |
| DT | 31.0 | - | 8.2 | - | 2.3 | - | 13.8 | - |
| BCQ | 49.1 | - | 17.1 | - | 30.8 | - | 32.3 | - |
| QDT | 57.3 | - | 13.3 | - | 31.0 | - | 33.9 | - |
| IQL | 42.1 | - | 34.9 | - | 61.7 | - | 46.2 | - |
| COMBO | 76.4 | - | 68.5 | - | 14.1 | - | 53.0 | - |
| TD3+BC | 14.8 | - | 62.1 | - | 88.6 | - | 55.2 | - |
| BEAR | 65.7 | - | 25.0 | - | 81.0 | - | 57.2 | - |
| BC | 88.9 | 14.6 | 38.3 | 16.3 | 1.5 | 17.1 | 42.9 | 16.0 |
| CQL | 94.7 | 37.1 | 41.8 | 32.1 | 49.6 | 29.6 | 62.0 | 32.9 |
| TT | 68.7 | 46.6 | 34.9 | 52.7 | 27.6 | 56.6 | 43.7 | 52.0 |
| SfBC | 73.9 | - | 73.8 | - | 74.4 | - | 74.0 | - |
| SynthER | 99.1 | - | 66.4 | - | 143.3 | - | 102.9 | - |
| Diffuser | 113.9 | - | 121.5 | - | 123.0 | - | 119.5 | - |
| HDMI | 120.1 | - | 121.8 | - | 128.6 | - | 123.5 | - |
| HD-DA | 72.8 | 45.5 | 42.1 | 54.7 | 80.7 | 45.7 | 65.2 | 48.6 |
| TCD | 128.1 | 29.8 | 132.9 | 41.4 | 146.4 | 75.5 | 135.8 | 48.9 |
| DD | 116.2 | 83.2 | 122.3 | 78.2 | 125.9 | 23.0 | 121.5 | 61.5 |
| AEPO | 136.0 | 107.2 | 128.4 | 109.9 | 132.4 | 165.5 | 132.3 | 127.5 |

| Environment | antmaze-umaze | | antmaze-medium | | antmaze-large | | mean score | total score |
|---|---|---|---|---|---|---|---|---|
| Environment type | | diverse | play | diverse | play | diverse | | |
| AWAC | 56.7 | 49.3 | 0.0 | 0.7 | 0.0 | 1.0 | 18.0 | 107.7 |
| DT | 59.2 | 53.0 | 0.0 | 0.0 | 0.0 | 0.0 | 18.7 | 112.2 |
| BC | 65.0 | 55.0 | 0.0 | 0.0 | 0.0 | 0.0 | 20.0 | 120.0 |
| BEAR | 73.0 | 61.0 | 0.0 | 8.0 | 0.0 | 0.0 | 23.7 | 142.0 |
| BCQ | 78.9 | 55.0 | 0.0 | 0.0 | 6.7 | 2.2 | 23.8 | 142.8 |
| TD3+BC | 78.6 | 71.4 | 10.6 | 3.0 | 0.2 | 0.0 | 27.3 | 163.8 |
| CQL | 74.0 | 84.0 | 61.2 | 53.7 | 15.8 | 14.9 | 50.6 | 303.6 |
| IQL | 87.5 | 62.2 | 71.2 | 70.0 | 39.6 | 47.5 | 63.0 | 378.0 |
| QDQ | 98.6 | 67.8 | 81.5 | 85.4 | 35.6 | 31.2 | 66.7 | 466.8 |
| DD | 73.1 | 49.2 | 0.0 | 24.6 | 0.0 | 7.5 | 25.7 | 154.4 |
| D-QL | 93.4 | 66.2 | 76.6 | 78.6 | 46.4 | 56.6 | 69.6 | 417.8 |
| IDQL | 93.8 | 62.0 | 86.6 | 83.5 | 57.0 | 56.4 | 73.2 | 439.3 |
| SfBC | 92.0 | 85.3 | 81.3 | 82.0 | 59.3 | 45.5 | 74.2 | 445.4 |
| QGPO | 96.4 | 74.4 | 83.6 | 83.8 | 66.6 | 64.8 | 78.3 | 469.6 |
| AEPO | 100.0 | 100.0 | 76.7 | 83.3 | 56.7 | 66.7 | 80.6 | 483.4 |

## 5.1 Environments

We select D4RL tasks [21] as the test bed, which contains four types of benchmarks, Gym-MuJoCo, Pointmaze, Locomotion, and Adroit, with different dataset qualities. Gym-MuJoCo tasks are composed of HalfCheetah, Hopper, and Walker2d with different difficulty settings datasets (e.g., medium, medium-replay, and medium-expert), where 'medium-replay' and 'medium-expert' denote the mixture level of behavior policies and 'medium' represents uni-level behavior policy. In Pointmaze, we select Maze2D with three difficulty settings (umaze, medium, and large) and two reward settings (sparse and dense) for evaluation. Locomotion contains an ant robot control task with different maze sizes, which we distinguish using umaze, medium, and large. These datasets include an abundant fraction of near-optimal episodes, making training challenging. Adroit contains several sparse-reward, high-dimensional hand manipulation tasks where the datasets are collected under three types of situations (human, expert, and cloned).

## 5.2 Metrics

Considering the various reward structures of different environments, we select the normalized score as the comparison metric. As an example, the normalized score $E_{norm}$ is calculated by $E_{norm} = \frac{E - E_{random}}{E_{expert} - E_{random}} * 100$, where $E_{random}$ and $E_{expert}$ are the performance of random and expert policies, and $E$ is the evaluation performance. In the ablation study, we select the radar chart

to compare the holistic capacity of all methods, where the normalized performance is calculated by $E_{norm} = \frac{E_i}{\max(\{E_j|j \in 1,...,N\})}$, where $E_i$ is the evaluation performance of method $i$, $i$ indicates the index of comparison methods, and $N$ is the total number of methods.

## 5.3 Baselines

For baselines, we compare our method with diffusion-based methods (DiffuserLite [19], HD-DA [10], IDQL [25], AdaptDiffuser [59], Consistency-AC [18], HDMI [57], TCD [31], QGPO [64], SfBC [11], DD [1], Diffuser [48], D-QL [82], and D-QL@1, etc.) and non-diffusion-based methods, including traditional RL methods (AWAC [70] and BC), model-based methods (TAP [49], MOReL [51], MOPO [89], and MBOP [3]), constraint-based methods (CQL [55], BCQ [23], and BEAR [54], etc.), uncertainty-based methods (PBRL [4], TD3+BC [22], and IQL [53], etc.), and transformer-based methods (BooT [80], TT [47], and DT [13], etc.). In summary, we compared more than 30 competitive methods across over 30 subtasks.

## 5.4 Results

In Table 1, we report the comparison results with dozens of diffusion-based and non-diffusion-based methods on the D4RL Gym-MuJoCo tasks. Among all the algorithms on 9 tasks, the best performance of our method (AEPO) illustrates the effectiveness in decision-making scenarios. Especially, in guided sampling, as a diffusion model, our method not only provides an approximate solution to guided sampling in theory but also surpasses most recent diffusion-based methods, such as QGPO, DiffuserLite, and D-QL, in abundant experiments. In order to validate the effectiveness in sparse and dense

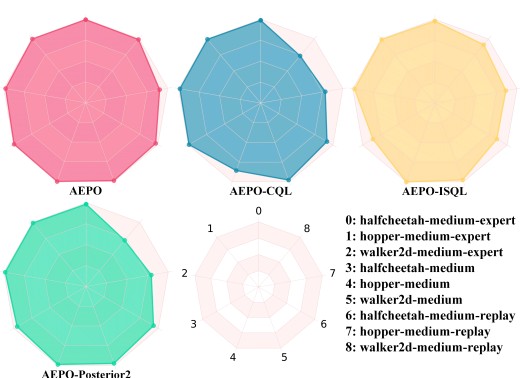

Figure 1: Q function training and posterior approximation ablation of AEPO on Gym-MuJoCo tasks.

reward settings, as well as hard tasks with the most sub-optimal trajectories, we conduct experiments on maze2d and antmaze tasks. The results are reported in Table 2, where we can see that our method approaches or surpasses the SOTA algorithms. Limited by the space, we postpone more experiments and discussion of D4RL Adroit in Table 3 of Appendix B.

Apart from Posterior 1 introduced in Section 4.2, we also investigate the performance of Posterior 2 on the D4RL Gym-MuJoCo tasks. To show the performance differences obviously, we adopt the holistic performance and show them on the polygon shown in Figure 1, where each vertex represents a sub-task. The fuller the polygon, the better the overall performance of the model. From the figure, it can be observed that AEPO outperforms AEPO-Posterior 2. This may be attributed to the variance calculation. As shown in Section 4.2, Posterior 2 is influenced by two factors—one being the intrinsic variance of the dataset and the other being the diffusion model.

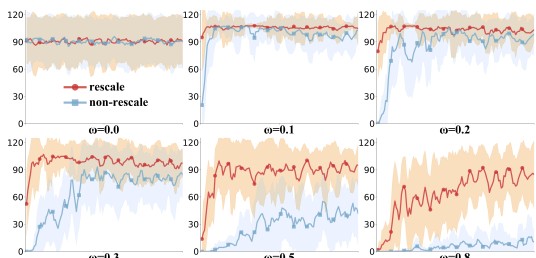

Figure 2: Guidance rescale ablation of AEPO on Gym-MuJoCo walker2d-medium-expert task. The x-axis denotes training steps.

In contrast, Posterior 1 is influenced solely by the diffusion model. For the different Q function training strategies, we also conduct experiments on the Gym-MuJoCo tasks. AEPO-CQL uses the contrastive q-learning to train the Q function, and AEPO-ISQL adopts in-support softmax q-learning to train the Q function. The results in Figure 1 illustrate that implicit Q-learning is an effective method to learn a better Q function in offline training.

As shown in Figure 2, guided sampling, such as $\omega = 0.1$, is important to reach higher returns compared with non-guided sampling ($\omega = 0$). With the increase of $\omega$, the performance of 'non-rescale' decreases quickly, while 'rescale' can still hold the performance. Obviously, guidance rescaling makes it robust to find better performance in a wider range of $\omega$.

# 6    Discussion

## 6.1    Differences Discussion

Previous methods, such as QGPO, cannot solve the intractable intermediate energy with the log-expectation form, which leads to sub-optimal performance. Facing this issue, we resolve the intractable log-expectation of intermediate energy with Taylor expansion and the moment generating function, achieving better performance in 33 tasks with various domains. Below we discuss the differences between our method and QGPO in detail.

- From the perspective of theory, our method provides a theoretical solution for the log-expectation of the intermediate energy, which takes a further step compared with QGPO. QGPO only derives the formula of intermediate energy rather than the further results of the log-expectation.
- From the perspective of algorithmic design, in Theorem 3.2 of QGPO, where the authors use contrastive learning loss to fit the intermediate energy, the authors mention that, under infinite data and model capacity, the contrastive loss (Equation (12) in QGPO) can perfectly approximate the intermediate energy. However, in practice, we cannot achieve infinite model capacity, and offline samples are certainly limited. Our method does not require assumptions about the sample size or model capacity.
- From the perspective of the dataset, QGPO requires the use of a diffusion model to generate fake action vectors for CEP, and these fake action vectors may introduce the influence of out-of-distribution actions. However, our method does not require the generation of fake actions.

Compared with posterior sampling methods, such as DPS, the key difference between DPS and AEPO is that the intermediate energy formula used in AEPO is exact, while that in DPS is inexact. The derivation of the exact and inexact intermediate energy can be found in Table 1 and Appendix E of QGPO. AEPO first bases on the exact intermediate energy, then solves the log-expectation that QGPO does not resolve.

## 6.2    Intuition Behind Using the Q-function as the Energy Function

In reinforcement learning, the Q-value $Q(s, a)$ represents the expected return of taking action $a$ in state $s$. Higher Q-values indicate more desirable actions. In energy-based guided sampling, the energy function $E(x)$ serves as a scoring function, where lower energy corresponds to higher sampling probability on $x$. By identifying the energy function as $E(x) = -Q(s, a)$, we make high-Q actions more likely to be sampled through the exponential weighting $\beta Q(s, a)$. This aligns with the optimization objective in constrained RL: maximizing the expected Q-value under a KL divergence constraint from a prior behavior policy.

# 7    Conclusion

In this paper, we provide a theoretical analysis of the intermediate energy that matters in conditional decision generation with diffusion models. We investigate the closed-form solution of intermediate guidance that has intractable log-expectation formulation and provide an effective approximation method under the most widely used Gaussian-based diffusion models. Finally, we conduct sufficient experiments in 4 types, 30+ tasks by comparing with 30+ baselines to validate the effectiveness. Limitations and potential improvements can be found in Appendix B.8.

## Acknowledgement

We would like to thank Lijun Bian for her contributions to the figures and tables of this manuscript. We thank Siyuan Guo for his contributions to the writing suggestions of this manuscript. This work is supported in part by the National Key R&D Program of China (No. 2023YFF0905400, No. 2021ZD0112500); the National Natural Science Foundation of China (No. 62476110, No. U2341229); the National Key R&D Program of China (No. 2023YFF0905400, No. 2021ZD0112500); the Key R&D Project of Jilin Province (No. 20240304200SF); NSFC Grant (No. 62576364).

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

## A   Pseudocode of AEPO

---

**Algorithm 1** Analytic Energy-guided Policy Optimization (AEPO).

---

 1: **Input:** Dataset $D_\mu$, Max iterations $M$ of training, environmental time limit $\mathcal{T}$, generation steps $T$, Noise prediction model $\epsilon_\theta$, intermediate energy model $\mathcal{E}_\Theta$, value function $V_\phi$, action-value function $Q_\psi$, target action-value function $Q_{\bar\psi}$
 2: **Output:** Well-trained $\epsilon_\theta$, $\mathcal{E}_\Theta$, $V_\phi$, $Q_\psi$
 3: // Training Process
 4: **for** $i = 1$ **to** $M$ **do**
 5:     // Train $\epsilon_\theta$ so that we can obtian $\nabla_{a_t} \log \mu_t(a_t|s) = -\epsilon_\theta(s, a_t, t)/\sigma_t$.
 6:     Train noise prediction model $\epsilon_\theta$ according to Equation (7)
 7:     // Train the Q function so that we can obtain $Q$ for any input $(s, a_0)$ and $Q'$ with autograd.
 8:     Train $V_\phi$ and $Q_\psi$ with loss $\mathcal{L}_V$ and $\mathcal{L}_Q$ (Equation (26))
 9:     // For each sampled data from the dataset, we calculate the approximate intermediate energy value with the posterior distribution.
10:     Calculate the target intermediate energy shown in Equation (24)
11:     // Use finite data point to fit intermediate energy function $\mathcal{E}_\Theta(s, a_t, t)$ with neural network.
12:     Update the parameter $\Theta$ according to Equation (25)
13: **end for**
14: // Evaluation Process
15: **for** $i = 1$ **to** $\mathcal{T}$ **do**
16:     Receive state $s$ from the environment
17:     Sample $a_T$ from $\mathcal{N}(0, \boldsymbol{I})$
18:     **for** $t = T$ **to** $0$ **do**
19:         Calculate $\nabla_{a_t} \log \mu_t(a_t|s)$ with Equation (13)
20:         Obtain $\nabla_{a_t} \mathcal{E}_\Theta(s, a_t, t)$ by performing gradient on $a_t$
21:         // $\nabla_{a_t} \log \pi_t(a_t|s) = \nabla_{a_t} \log \mu_t(a_t|s) + \nabla_{a_t} \mathcal{E}_\Theta(s, a_t, t)$
22:         Obtain score function $\nabla_{a_t} \log \pi_t(a_t|s)$ according to Equation (12)
23:         // The implementation follows DPM-solver to realize lower generation steps and reduce time consumption.
24:         Perform action denoising with Equation (6)
25:     **end for**
26:     Interact with the environment with generated action $a_0$
27: **end for**

---

The training and evaluation of AEPO are shown in Algorithm 1. In lines 3-13, we sample data from the dataset and train the noise prediction model $\epsilon_\theta$ that is used to obtain $\nabla_{a_t} \log \mu_t(a_t|s) = -\epsilon_\theta(s, a_t, t)/\sigma_t$, the intermediate energy $\mathcal{E}_\Theta(s, a_t, t)$ that can be used to approximate the intermediate guidance $\nabla_{a_t} \mathcal{E}_t(s, a_t)$, and the Q function $Q_\psi$ that is used to calculate $\mathcal{E}_\Theta(s, a_t, t)$. During the evaluation (lines 14-27), for each state received from the environment, we perform generation with Equation (6), where we use DPM-solver [62] as implementation. After several generation steps, we obtain the generated action $a_0$ to interact with the environment.

## B   Additional Experiments

### B.1   Additional Experiments on D4RL Adroit

D4RL Adroit is a hand-like robotic manipulation benchmark, which contains several sparse rewards and high-dimensional robotic manipulation tasks where the datasets are collected under three types of situations (-human, -expert, and -cloned) [73]. For example, the Pen is a scenario where the agent needs to get rewards by twirling a pen. The Relocate scenario requires the agent to pick up a ball and move it to a specific location. The experiments of D4RL Adroit are shown in Table 3, where we compare our method with non-diffusion-based methods highlighted with blue color and diffusion-based methods highlighted with red color. Our method is highlighted in yellow color. The results show that our method surpasses or matches the best performance in 11 of the total 12 subtasks, which illustrates strong competitiveness of our method.

Table 3: Offline RL algorithms comparison on Adroit. We select 4 tasks for evaluation, where each task contains 3 types of difficulty settings. We use red color , blue color , and yellow color to show diffusion-based baselines, non-diffusion-based baselines, and our method.

| Task | pen | | | hammer | | | door | | | relocate | | | mean score | total score |
|---|---|---|---|---|---|---|---|---|---|---|---|---|---|---|
| Dataset | human | expert | cloned | human | expert | cloned | human | expert | cloned | human | expert | cloned | | |
| BC | 7.5 | 69.7 | 6.6 | - | - | - | - | - | - | 0.1 | 57.1 | 0.1 | - | - |
| BEAR | -1.0 | - | 26.5 | - | - | - | - | - | - | - | - | - | - | - |
| BCQ | 68.9 | - | 44.0 | - | - | - | - | - | - | - | - | - | - | - |
| IQL | 71.5 | - | 37.3 | 1.4 | - | 2.1 | 4.3 | - | 1.6 | 0.1 | - | -0.2 | - | - |
| TT | 36.4 | 72.0 | 11.4 | 0.8 | 15.5 | 0.5 | 0.1 | 94.1 | -0.1 | 0.0 | 10.3 | -0.1 | 20.1 | 240.9 |
| CQL | 37.5 | 107.0 | 39.2 | 4.4 | 86.7 | 2.1 | 9.9 | 101.5 | 0.4 | 0.2 | 95.0 | -0.1 | 40.3 | 483.8 |
| UWAC | 65.0 | 119.8 | 45.1 | 8.3 | 128.8 | 1.2 | 10.7 | 105.4 | 1.2 | 0.5 | 108.7 | 0.0 | 49.6 | 594.7 |
| TAP | 76.5 | 127.4 | 57.4 | 1.4 | 127.6 | 1.2 | 8.8 | 104.8 | 11.7 | 0.2 | 105.8 | -0.2 | 51.9 | 622.6 |
| TCD | 49.9 | 35.6 | 73.3 | - | - | - | - | - | - | 0.4 | 59.6 | 0.2 | - | - |
| HD-DA | -2.6 | 107.9 | -2.7 | - | - | - | - | - | - | 0.0 | -0.1 | -0.2 | - | - |
| DiffuserLite | 33.2 | 20.7 | 2.1 | - | - | - | - | - | - | 0.1 | 0.1 | -0.2 | - | - |
| DD | 64.1 | 107.6 | 47.7 | 1.0 | 106.7 | 0.9 | 6.9 | 87.0 | 9.0 | 0.2 | 87.5 | -0.2 | 43.2 | 518.4 |
| HDMI | 66.2 | 109.5 | 48.3 | 1.2 | 111.8 | 1.0 | 7.1 | 85.9 | 9.3 | 0.1 | 91.3 | -0.1 | 44.3 | 531.6 |
| D-QL@1 | 66.0 | 112.6 | 49.3 | 1.3 | 114.8 | 1.1 | 8.0 | 93.7 | 10.6 | 0.2 | 95.2 | -0.2 | 46.1 | 552.6 |
| QGPO | 73.9 | 119.1 | 54.2 | 1.4 | 123.2 | 1.1 | 8.5 | 98.8 | 11.2 | 0.2 | 102.5 | -0.2 | 49.5 | 593.9 |
| LD | 79.0 | 131.2 | 60.7 | 4.6 | 132.5 | 4.2 | 9.8 | 111.9 | 12.0 | 0.2 | 109.5 | -0.1 | 54.6 | 655.5 |
| AEPO | 76.7 | 147.0 | 69.3 | 10.3 | 129.7 | 6.4 | 9.0 | 106.5 | 3.9 | 0.8 | 107.0 | 0.6 | 55.6 | 667.5 |

Table 4: The best performance comparison of guidance rescale and non-guidance rescale with grid search.

| Type | guidance rescale | non-guidance rescale |
|---|---|---|
| walker2d-me | $109.3_{\pm 0.5}$ | $109.4_{\pm 0.9}$ |
| walker2d-mr | $90.8_{\pm 1.5}$ | $92.8_{\pm 4.8}$ |
| halfcheetah-mr | $43.7_{\pm 1.3}$ | $43.3_{\pm 1.5}$ |

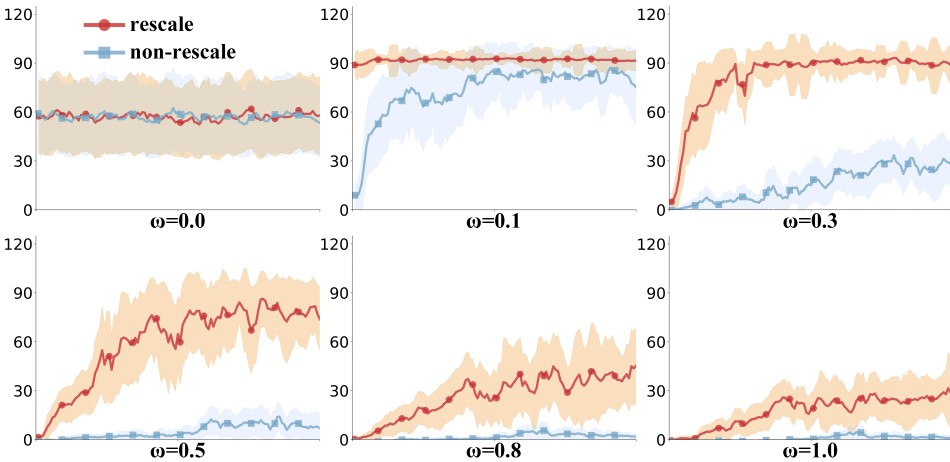

Figure 3: Guidance rescale ablation of AEPO on D4RL Gym-MuJoCo halfcheetah-medium-expert task. The y-axis and x-axis denote the normalized score and training steps, respectively.

## B.2 Additional Rescaling Ablation

We want to emphasize that the purpose of using guidance rescale is to make our method less sensitive to $\omega$. The main contributions of our method are our theoretical results of intermediate energy and systematic comparison results with dozens of baselines on 33 tasks. The results in Table 4 show that the best performance achieved via grid search under both guidance rescale and non-guidance rescale settings is comparable, indicating that guidance rescale is not the primary factor contributing to significant performance gains.

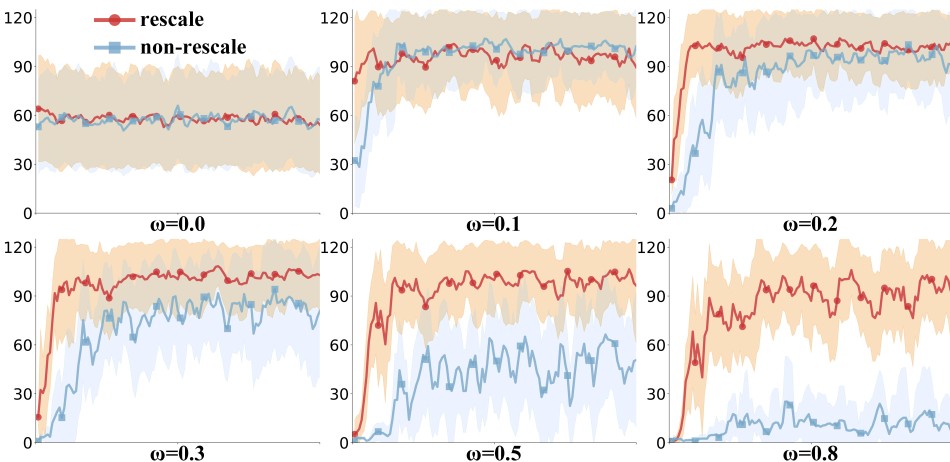

Figure 4: Guidance rescale ablation of AEPO on D4RL Gym-MuJoCo hopper-medium-expert task. The y-axis and x-axis denote the normalized score and training steps, respectively.

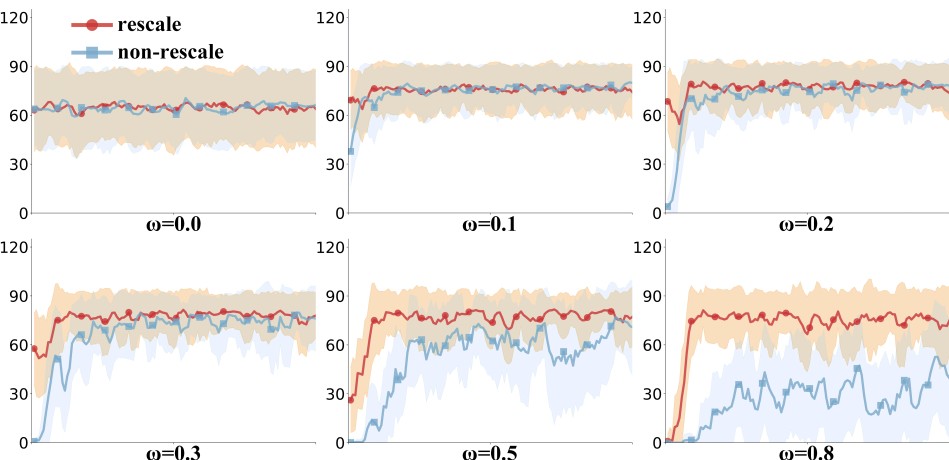

Figure 5: Guidance rescale ablation of AEPO on D4RL Gym-MuJoCo walker2d-medium task. The y-axis and x-axis denote the normalized score and training steps, respectively.

Apart from the guidance rescaling ablation on walker2d-medium-expert task, we also conduct the experiments on halfcheetach-medium-expert, hopper-medium-expert, and walker2d-medium tasks, where the results are shown in Figure 3, Figure 4, and Figure 5, respectively. From the results, we can see that guidance is important to reach a better performance by comparing $\omega = 0.1$ and $\omega = 0$, where $\omega = 0$ means no intermediate guidance. 'rescale' means we apply the guidance rescaling strategy, and 'non-rescale' means we do not use guidance rescaling. With the increase of $\omega$, the performance of 'non-rescale' decreases quickly, while 'rescale' can still hold the performance, which illustrates that guidance rescaling makes the model robust to the guidance degree $\omega$, thus leading to better performance in a wider range of $\omega$.

## B.3 Parameter Sensitivity

We investigate the influence of hyperparameters of $\beta$ and $\tau$ that are used in loss $\mathcal{L}_{IE}$ (Equation (25)) and $\mathcal{L}_Q$ (Equation (26)). From the results shown in Figure 6, we can see that our method shows slight sensitivity to $\beta$ value. For the hyperparameter $\tau$, we find that a certain value is friendly for achieving good performance, such as $\tau = 0.7$ (Figure 6 (g)).

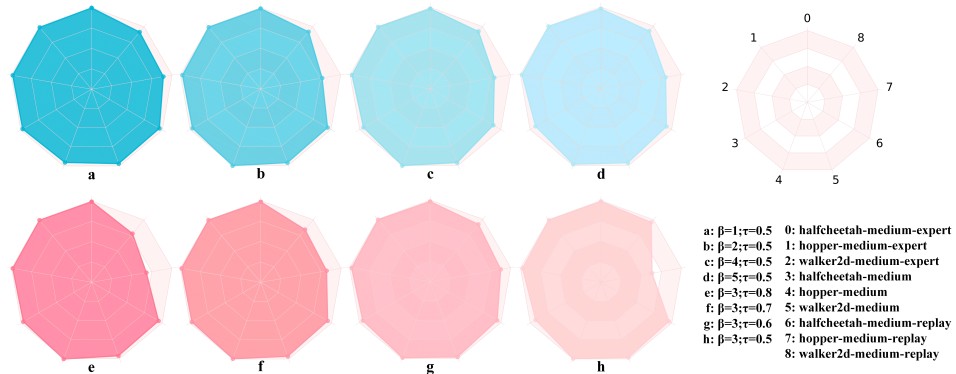

Figure 6: Parameter sensitivity of D4RL Gym-MuJoCo tasks.

Table 5: The hyperparameters of AEPO.

| Hyperparameter | Value |
| --- | --- |
| network backbone | MLP |
| action value function ($Q_\psi$) hidden layer | 3 |
| action value function ($Q_\psi$) hidden layer neuron | 256 |
| state value function ($V_\phi$) hidden layer | 3 |
| state value function ($V_\phi$) hidden layer neuron | 256 |
| intermediate energy function ($\mathcal{E}_\Theta$) hidden layer | 3 |
| intermediate energy function ($\mathcal{E}_\Theta$) hidden layer neuron | 256 |
| noise prediction function ($\epsilon_\theta$) hidden block | 7 |
| noise prediction function ($\epsilon_\theta$) hidden layer neuron | 256/512/1024 |
| inverse temperature $\beta$ | 1 |
| expectile weight $\tau$ | 0.5 |
| guidance degree $\omega$ | 0.1 |
| $\nu$ | 0.001 |
| $\bar{a}, u_0$ | Refer to Appendix K.2 |

## B.4 Computation

We conduct the experiments on NVIDIA GeForce RTX 3090 GPUs and NVIDIA A10 GPUs, and the CPU type is Intel(R) Xeon(R) Gold 6230 CPU @ 2.10GHz. Each run of the experiments spanned about 48-72 hours, depending on different tasks.

## B.5 Hyperparameters

The hyperparameters used in our method are shown in Table 5.

## B.6 Computational Efficiency

We select hopper-m as the test environment, set the batch size to 32, and control the same hardware conditions. The GPU is NVIDIA A10, and the CPU is Intel(R) Xeon(R) Gold 6230 CPU @ 2.10GHz. The results are shown in Table 6, where we report the results of runtime GPU memory usage, inference time consumption of every decision, and training time consumption of every neural network update. The results indicate that our method incurs lower GPU memory overhead compared to QGPO, primarily because it does not require loading additional 'fake_actions' data. In terms of training time, our method introduces higher overhead than QGPO, mainly due to the extra computation involved in calculating the intermediate energy of Equation (16). However, in inference time, the cost of AEPO and QGPO is comparable, as both adopt the same number of generation steps.

Table 6: The computational efficiency comparison between AEPO and the SOTA diffusion-based model QGPO. We set the batch size to 32 and conduct experiments of hopper-m under the same hardware conditions. The GPU used is NVIDIA A10, and the CPU is Intel(R) Xeon(R) Gold 6230 CPU @ 2.10GHz.

| Method | AEPO | QGPO |
|---|---|---|
| training time of every updating on hopper-m (s) | 0.018 | 0.015 |
| GPU memory usage with batch size 32 on hopper-m (M) | 1135 | 1161 |
| inference time of every generation on hopper-m (s) | 0.068 | 0.070 |

Table 7: The influence of different training methods on the Q function. The experiments are conducted on AEPO, but with different training methods of Q functions.

| Q function training type | IQL-like Q-function training | CQL-like Q-function training |
|---|---|---|
| walker2d-me | $109.3_{\pm 0.5}$ | $109.8_{\pm 0.7}$ |
| halfcheetah-me | $94.4_{\pm 0.9}$ | $93.8_{\pm 0.9}$ |
| hopper-me | $111.5_{\pm 1.1}$ | $111.2_{\pm 2.4}$ |

### B.7 The Effects of Q-function Training Method

Regardless of which Q-function training method is used, as long as the Q-function learning is suitable for offline RL, the performance obtained using our method is similar. For example, with IQL-like Q function training methods and CQL-like Q function training methods, when obtaining accurately estimated Q functions, the performance of AEPO is similar. As shown by the experimental results in Table 7, this experiment used two methods to train the Q function, one being an IQL-like training method and the other using a CQL-like training method. We find that the experimental performance was similar, indicating that we just need to adopt the Q-function training method that is suitable and widely used in offline RL.

### B.8 Limitation Discussion

For the limitation of this paper, we summarize the potential areas for improvement as follows.

- The generative model requires multiple iterative steps for each decision, which may impact its deployment in environments with strict real-time requirements.
- The posterior estimation method proposed in this paper is sampling-based. We could also consider higher-order posterior estimation methods in the future, such as the second-order Tweedie method [75]. However, this would require a trade-off between more accurate posterior estimation and increased computational cost.
- The current method is primarily designed for continuous action spaces, where the use of the first-order Taylor expansion of the Q-function with respect to actions is mathematically justified due to the differentiability of Q(s, a) with respect to the continuous action variable.

## C  The Constrained RL Problem

The constrained RL problem is

$$\min_{\pi} \quad - \mathbb{E}_{s \sim D^{\mu}} \left[ \mathbb{E}_{a \sim \pi(\cdot|s)} Q(s, a) \right]$$
$$s.t. \quad D_{KL}(\pi(\cdot|s)||\mu(\cdot|s)) \tag{27}$$

which can be converted to

$$\max_{\pi} \mathbb{E}_{s \sim D^{\mu}} \left[ \mathbb{E}_{a \sim \pi(\cdot|s)} Q(s, a) - \frac{1}{\beta} D_{KL}(\pi(\cdot|s)||\mu(\cdot|s)) \right] . \tag{28}$$

It reveals the optimal solution

$$\pi^*(a|s) \propto \mu(a|s) e^{\beta Q(s,a)},$$

where $\pi^*(a|s)$ is the optimal policy that can produce optimal decisions by learning from the dataset, $\mu(a|s)$ is the behavior policy that is used to sample the data of the dataset, i.e., the data distribution of the dataset induced from $\mu(a|s)$, $Q(s,a)$ is the action value function, and we usually adopt neural network to learn the action value function.

*Proof.* The Equation (28) is actually the lagrangian function of the contained RL problem by selecting the lagrangian multiplier as $\frac{1}{\beta}$ and adding the implicit constraint $\int \pi(a|s)da = 1$

$$\min_{\pi} \ \mathbb{E}_{s \sim D^{\mu}} \mathcal{L}$$

$$= \min_{\pi} \ \mathbb{E}_{s \sim D^{\mu}} \left[ \mathbb{E}_{a \sim \pi(\cdot|s)} Q(s,a) - \frac{1}{\beta} * D_{KL}(\pi(\cdot|s)||\mu(\cdot|s)) + \eta * \left( \int \pi(a|s)da - 1 \right) \right].$$

The functional derivative of $\mathcal{L}$ regarding $\pi$ is

$$\frac{\partial \mathcal{L}}{\partial \pi} = \frac{\mathcal{L}(\pi + \delta\pi) - \mathcal{L}(\pi)}{\partial \pi}$$

$$\mathcal{L}(\pi + \delta\pi) = \int (\pi(a|s) + \delta\pi(a|s))Q_{\psi}(s,a)da$$

$$- \frac{1}{\beta} * \int (\pi(a|s) + \delta\pi(a|s)) \log \frac{\pi(a|s) + \delta\pi(a|s)}{\mu(\cdot|s)} da$$

$$+ \eta * \left( \int (\pi(a|s) + \delta\pi(a|s))da - 1 \right).$$

Notice that the Taylor expansion of $\log \frac{u(x)}{q(x)}$ at $u(x) = p(x)$ is

$$\log \frac{u(x)}{q(x)}\Big|_{u(x)=p(x)} = \log \frac{p(x)}{q(x)}$$

$$\left( \log \frac{u(x)}{q(x)} \right)' \Big|_{u(x)=p(x)} = \frac{1}{p(x)}$$

Thus,

$$\log \frac{u(x)}{q(x)} \approx \log \frac{u(x)}{q(x)}\Big|_{u(x)=p(x)} + \left( \log \frac{u(x)}{q(x)} \right)' \Big|_{u(x)=p(x)} (u(x) - p(x))$$

Let $u(x) = \pi(a|s) + \delta\pi(a|s), p(x) = \pi(a|s), q(x) = \mu(a|s)$, we have

$$\log \frac{\pi(a|s) + \delta\pi(a|s)}{\mu(\cdot|s)} \approx \log \frac{\pi(a|s)}{\mu(a|s)} + \frac{1}{\pi(a|s)} \delta\pi(a|s).$$

Then, $\mathcal{L}(\pi + \delta\pi)$ can be simplified

$$\mathcal{L}(\pi + \delta\pi) = \int (\pi(a|s) + \delta\pi(a|s))Q(s,a)da$$

$$- \frac{1}{\beta} * \int (\pi(a|s) + \delta\pi(a|s)) \left( \log \frac{\pi(a|s)}{\mu(a|s)} + \frac{1}{\pi(a|s)} \delta\pi(a|s) \right) da$$

$$+ \eta * \left( \int (\pi(a|s) + \delta\pi(a|s))da - 1 \right).$$

$$= \int \pi(a|s)Q(s,a)da - \frac{1}{\beta} * \int \pi(a|s) \log \frac{\pi(a|s)}{\mu(a|s)} da + \eta * \left( \int \pi(a|s)da - 1 \right)$$

$$+ \int \delta\pi(a|s) \left[ Q(s,a) - \frac{1}{\beta} \left( \log \frac{\pi(a|s)}{\mu(a|s)} + 1 \right) + \eta \right] - \frac{1}{\beta} \frac{1}{\pi(a|s)} (\delta\pi(a|s))^2$$

$$\approx \mathcal{L}(\pi) + \int \delta\pi(a|s) \left[ Q(s,a) - \frac{1}{\beta} \left( \log \frac{\pi(a|s)}{\mu(a|s)} + 1 \right) + \eta \right] da$$

Finally, we obtain the functional derivative is

$$\frac{\partial \mathcal{L}}{\partial \pi} = \frac{\mathcal{L}(\pi + \delta\pi) - \mathcal{L}(\pi)}{\partial \pi}$$

$$= Q(s,a) - \frac{1}{\beta}\left(\log \frac{\pi(a|s)}{\mu(a|s)} + 1\right) + \eta$$

Let $\frac{\partial \mathcal{L}}{\partial \pi} = 0$, we have

$$\pi^*(a|s) = \mu(a|s) * e^{\beta(Q(s,a)+\eta)-1}$$

$$\int \pi^*(a|s)da = \int \mu(a|s) * e^{\beta(Q(s,a)+\eta)-1}da = 1$$

$$\int \mu(a|s) * e^{\beta*Q(s,a)}da = e^{\beta*\eta-1}$$

$$\pi^*(a|s) = \frac{\mu(a|s) * e^{\beta*Q(s,a)}}{\int \mu(a|s) * e^{\beta*Q(s,a)}da}$$

$$\pi^*(a|s) \propto \mu(a|s) * e^{\beta*Q(s,a)}$$

$\square$

## D   Analysis of Exact and Inexact Intermediate Guidance

Previous studies propose that through an ingenious definition (For clarity, we rewrite it below), we can guarantee the relation $p_t(x_t) \propto q_t(x_t)e^{-\mathcal{E}_t(x_t)}$ of $p(x_t)$ and $q(x_t)$ at any time $t$, where $\mathcal{E}_t(x_t)$ is general formula of intermediate energy.

**Theorem D.1** (Intermediate Energy Guidance). *Suppose $p_0(x_0)$ and $q_0(x_0)$ has the relation of Equation (2). For $t \in (0, T]$, let $p_{t|0}(x_t|x_0)$ and $q_{t|0}(x_t|x_0)$ be defined by*

$$p_{t|0}(x_t|x_0) \coloneqq q_{t|0}(x_t|x_0) = \mathcal{N}(x_t; \alpha_t x_0, \sigma_t^2 \mathbf{I}).$$

*So the marginal distribution $p_t(x_t)$ and $q_t(x_t)$ at time $t$ are $p_t(x_t) = \int p_{t|0}(x_t|x_0)p_0(x_0)dx_0$ and $q_t(x_t) = \int q_{t|0}(x_t|x_0)q_0(x_0)dx_0$. Define a general representation of intermediate energy as*

$$\mathcal{E}_t(x_t) = \begin{cases} \beta\mathcal{E}(x_0), & t = 0 \\ -\log \mathbb{E}_{q_{0|t}(x_0|x_t)}[e^{-\beta\mathcal{E}(x_0)}], & t > 0 \end{cases} \tag{29}$$

*Then $q_t(x_t)$ and $p_t(x_t)$ satisfy*

$$p_t(x_t) \propto q_t(x_t)e^{-\mathcal{E}_t(x_t)} \tag{30}$$

*for any diffusion step $t$ and the score functions of $q_t(x_t)$ and $p_t(x_t)$ satisfy $\nabla_{x_t} \log p_t(x_t) = \nabla_{x_t} \log q_t(x_t) - \nabla_x \mathcal{E}_t(x_t)$.*

The derivation can be found in Appendix E.

While for inexact intermediate energy

$$\mathcal{E}_t^{MSE}(x_t) = \mathbb{E}_{q_{0|t}(x_0|x_t)}[\mathcal{E}(x_0)], t > 0,$$

$$\mathcal{E}_t^{DPS}(x_t) = \mathcal{E}(\mathbb{E}_{q_{0|t}(x_0|x_t)}[x_0]), t > 0,$$

that are adopted in classifier-guided and classifier-free-guided methods, it can be derivated that $\mathcal{E}_t(x_t) \geq \mathcal{E}_t^{MSE}(x_t)$ when $\beta = 1$ and $t > 0$.

*Proof.* By applying Jensen's Inequality, it is easy to have

$$-\log \mathbb{E}_{q_{0|t}(x_0|x_t)}[e^{-\beta\mathcal{E}(x_0)}] \geq -\mathbb{E}_{q_{0|t}(x_0|x_t)}[\log e^{-\beta\mathcal{E}(x_0)}] = \beta\mathbb{E}_{q_{0|t}(x_0|x_t)}[\mathcal{E}(x_0)].$$

When $\beta = 1$, we have $\mathcal{E}_t(x_t) \geq \mathcal{E}_t^{MSE}(x_t)$ $\square$

Also, we can prove that these three intermediate guidance values are not equal to each other:
$\nabla_{x_t}\mathcal{E}_t(x_t) \neq \nabla_{x_t}\mathcal{E}_t^{MSE}(x_t) \neq \nabla_{x_t}\mathcal{E}_t^{DPS}(x_t)$.

*Proof.* For $t > 0$,

$$\nabla_{x_t}\mathcal{E}_t(x_t) = \nabla_{x_t} - \log \mathbb{E}_{q_{0|t}(x_0|x_t)}[e^{-\beta\mathcal{E}(x_0)}],$$

$$= -\frac{1}{\mathbb{E}_{q_{0|t}(x_0|x_t)}[e^{-\beta\mathcal{E}(x_0)}]}\nabla_{x_t}\int q_{0|t}(x_0|x_t)[e^{-\beta\mathcal{E}(x_0)}]dx_0,$$

$$= -e^{\mathcal{E}_t(x_t)}\int q_{0|t}(x_0|x_t)\nabla_{x_t}\log q_{0|t}(x_0|x_t)e^{-\beta\mathcal{E}(x_0)}dx_0,$$

$$= -\mathbb{E}_{q_{0|t}(x_0|x_t)}\left[e^{\mathcal{E}_t(x_t)-\beta\mathcal{E}(x_0)}\nabla_{x_t}\log q_{0|t}(x_0|x_t)\right],$$

where we use the chain rule of derivation and gradient trick to obtain the result. when $\beta = 1$, we have $\nabla_{x_t}\mathcal{E}_t(x_t) = -\mathbb{E}_{q_{0|t}(x_0|x_t)}\left[e^{\mathcal{E}_t(x_t)-\mathcal{E}(x_0)}\nabla_{x_t}\log q_{0|t}(x_0|x_t)\right]$. Similarly, we can obtain the gradient of $\mathcal{E}_t^{MSE}(x_t)$ and $\mathcal{E}_t^{DPS}(x_t)$:

$$\nabla_{x_t}\mathcal{E}_t^{MSE}(x_t) = \nabla_{x_t}\mathbb{E}_{q_{0|t}(x_0|x_t)}[\mathcal{E}(x_0)],$$

$$= \nabla_{x_t}\int q_{0|t}(x_0|x_t)\mathcal{E}(x_0)dx_0,$$

$$= \int q_{0|t}(x_0|x_t)\nabla_{x_t}\log q_{0|t}(x_0|x_t)\mathcal{E}(x_0)dx_0,$$

$$= \mathbb{E}_{q_{0|t}(x_0|x_t)}\left[\mathcal{E}(x_0)\nabla_{x_t}\log q_{0|t}(x_0|x_t)\right].$$

$$\nabla_{x_t}\mathcal{E}_t^{DPS}(x_t) = \nabla_{x_t}\mathcal{E}(y)|_{y=\mathbb{E}_{q_{0|t}(x_0|x_t)}[x_0]},$$

$$= \nabla_y\mathcal{E}(y)^\top\nabla_{x_t}\mathbb{E}_{q_{0|t}(x_0|x_t)}[x_0],$$

$$= \nabla_y\mathcal{E}(y)^\top\mathbb{E}_{q_{0|t}(x_0|x_t)}\left[x_0\nabla_{x_t}\log q_{0|t}(x_0|x_t)\right],$$

$$= \mathbb{E}_{q_{0|t}(x_0|x_t)}\left[\nabla_y\mathcal{E}(y)^\top x_0\nabla_{x_t}\log q_{0|t}(x_0|x_t)\right].$$

From the results we can see that $\nabla_{x_t}\mathcal{E}_t(x_t) \neq \nabla_{x_t}\mathcal{E}_t^{MSE}(x_t) \neq \nabla_{x_t}\mathcal{E}_t^{DPS}(x_t)$. $\square$

## E   Guidance of Intermediate Diffusion Steps

In order to guide the diffusion model during the intermediate diffusion steps, we first consider the following problem

$$\min_p \mathbb{E}_{x_0\sim p_0(x_0)}\mathcal{E}(x_0) \tag{31}$$
$$s.t. \ D_{KL}(p_0(x_0)||q_0(x_0))$$

and the relation function

$$p_0(x_0) \propto q_0(x_0)e^{-\beta\mathcal{E}(x_0)} \tag{32}$$

Define

$$p_{t|0}(x_t|x_0) := q_{t|0}(x_t|x_0) = \mathcal{N}(x_t; \alpha_t x_0, \sigma_t^2 \boldsymbol{I})$$

$$p_t(x_t) = \int p_{t|0}(x_t|x_0)p_0(x_0)dx_0,$$

$$q_t(x_t) = \int q_{t|0}(x_t|x_0)q_0(x_0)dx_0,$$

$$\mathcal{E}_t(x_t) = \begin{cases} \beta\mathcal{E}(x_0), & t = 0 \\ -\log \mathbb{E}_{q_{0|t}(x_0|x_t)}[e^{-\beta\mathcal{E}(x_0)}], & t > 0 \end{cases}$$

Then, we will obtain the following relation between score function of $p_t(x_t)$ and the score function of of $q_t(x_t)$

$$\nabla_{x_t}\log p_t(x_t) = \nabla_{x_t}q_t(x_t) - \nabla_{x_t}\mathcal{E}_t(x_t) \tag{33}$$

for each intermediate diffusion step.

*Proof.* Known that

$$p_{t|0}(x_t|x_0) = q_{t|0}(x_t|x_0) = \mathcal{N}(x_t|\alpha_t x_0, \sigma_t^2 \boldsymbol{I})$$

$$q_t(x_t) = \int q_{t|0}(x_t|x_0) q_0(x_0) dx_0$$

$$p_t(x_t) = \int p_{t|0}(x_t|x_0) p_0(x_0) dx_0$$

$$p_0(x_0) \propto q_0(x_0) e^{-\beta \mathcal{E}(x_0)}$$

By integral

$$Z = \int q_0(x_0) e^{-\beta \mathcal{E}(x_0)} dx_0 = \mathbb{E}_{q_0(x_0)} \left[ e^{-\beta \mathcal{E}(x_0)} \right],$$

and

$$p_0(x_0) \propto q_0(x_0) e^{-\beta \mathcal{E}(x_0)},$$

we know

$$p_0(x_0) = \frac{q_0(x_0) e^{-\beta \mathcal{E}(x_0)}}{Z}.$$

Thus,

$$\begin{aligned}
p_t(x_t) &= \int p_{t|0}(x_t|x_0) p_0(x_0) dx_0 \\
&= \int p_{t|0}(x_t|x_0) \frac{q_0(x_0) e^{-\beta \mathcal{E}(x_0)}}{Z} dx_0 \\
&= \int q_{t|0}(x_t|x_0) \frac{q_0(x_0) e^{-\beta \mathcal{E}(x_0)}}{Z} dx_0 \\
&= q_t(x_t) \int q_{0|t}(x_0|x_t) \frac{e^{-\beta \mathcal{E}(x_0)}}{Z} dx_0 \\
&= \frac{q_t(x_t) \mathbb{E}_{q_{0|t}(x_0|x_t)}[e^{-\beta \mathcal{E}(x_0)}]}{Z} \\
&= \frac{q_t(x_t) e^{-\mathcal{E}_t(x_t)}}{Z} \\
p_t(x_t) &\propto q_t(x_t) e^{-\mathcal{E}_t(x_t)},
\end{aligned}$$

$\square$

where we replace $q_{t|0}(x_t|x_0) q_0(x_0)$ with $q_t(x_t) q_{0|t}(x_0|x_t)$ by using Bayes Law in the fourth equation. Now, if we want to generate data from $p(x_0)$, we just need train a diffusion model $q(x_0)$ and a guidance $\mathcal{E}_t$, then use

$$\begin{aligned}
p_t(x_t) &\propto q_t(x_t) e^{-\mathcal{E}_t(x_t)} \\
\nabla_{x_t} \log p_t(x_t) &= \nabla_{x_t} \log q_t(x_t) - \nabla_{x_t} \mathcal{E}_t(x_t)
\end{aligned} \tag{34}$$

to obtain the score function value of $\nabla_{x_t} \log p_t(x_t)$. Then, we can use the reverse transformation (Equation (6)) to generate samples.

In RL, we should use $Q(s, a)$ to denote $-\mathcal{E}(x_0)$, because Equation (1) and Equation (3) indicate that maximizing $Q(s, a)$ is same with minimizing $\mathcal{E}_0(x_0)$. Besides, we notice that approximating $\mathcal{E}_t(x_t)$ needs to calculate $\mathcal{E}(x_0)$. And actually $\mathcal{E}(x_0)$ is the $Q(s, a)$. In other words, the intermediate energy guidance in RL is defined by

$$\mathcal{E}_t(x_t) = \begin{cases} \beta \mathcal{E}(x_0) \to \beta Q(s, a), & t = 0 \\ -\log \mathbb{E}_{q_{0|t}(x_0|x_t)}[e^{-\beta \mathcal{E}(x_0)}] \to \log \mathbb{E}_{q_{0|t}(x_0|x_t)}[e^{\beta Q(s,a)}], & t > 0 \end{cases} \tag{35}$$

*Proof.*

$$\pi_t(a_t|s) = \int \pi_{t|0}(a_t|a_0, s)\pi_0(a_0|s)da_0$$

$$= \int \pi_{t|0}(a_t|a_0, s)\frac{\mu_0(a_0|s)e^{\beta Q(s,a_0)}}{Z}da_0$$

$$= \int \pi_{t|0}(a_t|a_0, s)\frac{\mu_0(a_0|s)e^{\beta Q(s,a_0)}}{Z}da_0$$

$$= \mu_t(a_t|s) \int \mu_0(a_0|a_t, s)\frac{e^{\beta Q(s,a_0)}}{Z}da_0$$

$$= \frac{\mu_t(a_t|s)\mathbb{E}_{\mu_0(a_0|a_t,s)}[e^{\beta Q(s,a_0)}]}{Z}$$

$$= \frac{\mu_t(a_t|s)e^{\mathcal{E}_t(a_t)}}{Z}$$

$$\pi_t(a_t|s) \propto \mu_t(a_t|s)e^{\mathcal{E}_t(s,a_t)}$$

$\square$

Here, we slightly abuse the notion of $\mathcal{E}$ because, in RL, the input of the energy function is the concatenation of state and action.

## F   Discussion of Previous Intermediate Energy Estimation Methods

Here we discuss the differences between our method (AEPO) and QGPO [64] and Diffusion-DICE [67] from several perspectives.

- From a theoretical perspective: Our method provides a theoretical solution for the log expectation of the intermediate energy, whereas QGPO and Diffusion-DICE do not discuss further results for the log expectation term. In Lemma 1 of the Diffusion-DICE paper, it reformulates the estimation of the log expectation term of the intermediate energy as a convex optimization problem, which is different from our approach: we aim to derive further analytical results for the log expectation term.

- From the action generation perspective: When generating actions, Diffusion-DICE requires generating multiple candidate actions (Algorithm 1, line 10 in Diffusion-DICE), whereas our method, like QGPO, does not require generating multiple candidates.

- For additional assumptions: In Theorem 3.2 of QGPO where the authors use contrastive learning loss to fit the intermediate energy, the authors mention that, under infinite data and model capacity, the contrastive loss (Equation (12) in QGPO) can perfectly approximate the intermediate energy. However, in practice, we cannot achieve infinite model capacity, and offline samples are certainly limited. Our method does not require assumptions about the sample size or model capacity.

- For the dataset: Furthermore, QGPO requires the use of a diffusion model to generate fake action vectors for CEP, and these fake action vectors may introduce the influence of out-of-distribution actions. However, our method does not require the generation of fake actions.

- For training stability:   Diffusion-DICE formulates an optimization problem as $\min \mathbb{E}[f(x)e^{-y} + y]$, where $f(x)$ represents the Q function in RL, is usually positive. This results in an adversarial training dynamic: the term $\min e^{-y}$ encourages larger values of $y$, while $\min y$ pushes the output $y$ lower. However, when $y > 0$, the gradient of $e^{-y}$ dominates that of $y$, which can drive the model to continuously increase its output prediction, eventually leading to unstable training. This concern is also raised by readers in the GitHub issues of Diffusion-DICE. Fortunately, this issue does not arise in our method (AEPO) and QGPO. Our method removes the log and exp operations through simplification and approximation, and QGPO stabilizes training by normalizing weights during each CEP update.

Table 8: The intermediate energies comparison between QGPO and AEPO. We select 'hopper-medium-v2' as the evaluation task, where the reference values are calculated with the analytic equation results (Equation (24)). The estimated intermediate values of AEPO and QGPO are calculated with the learned neural network $\mathcal{E}_\Theta$ and the $f_\phi$ trained with CEP, respectively.

| method | AEPO | QGPO |
|---|---|---|
| mean error | 48.8 | 53.2 |
| performance | 87.4 | 86.6 |

- Experiments of the estimated intermediate energies: The quality of the intermediate energy estimation is directly related to the model's performance. During the action generation process, if the estimation of the intermediate energy is biased, the generated actions are unlikely to be good, leading to inferior evaluation performance. We report the mean error between the estimated intermediate energies from AEPO and QGPO and the calculated value from Equation (16). The results are shown in Table 8.

## G  Discussion of the Q-function Smoothness and Locality

In Equation 14, we use the Taylor expansion to convert the implicit dependence on the action in the exponential term to explicit dependence. Below we provide the necessary conditions analysis of Q-function smoothness and the locality when applying the Taylor expansion on Q-function.

For smoothness, we often use neural networks to learn the Q function in practical implementation, and by using specific activation functions, such as SiLU (the activation function used in our source code), we can ensure that the Q function is smooth with respect to actions and at least twice differentiable. This is because: If the activation function $\phi(\cdot)$ is SiLU, then $\phi(\cdot)$ is infinitely differentiable with respect to the input. Assuming the neural network is an L-layer feedforward network, i.e., $Q(a) = W_L \phi(W_{L-1} \phi(\ldots))$, then 1) affine transformations are infinitely differentiable, 2) activation functions are infinitely differentiable, so the composite function $Q(a)$ with respect to $a$ is also infinitely differentiable, and at least twice differentiable. Even in cases where activation functions like ReLU are used, the Q-function remains piecewise linear and locally approximable by linear functions in regions away from non-differentiable points.

For local linearity, since the Hessian of $Q(a)$ with respect to a exists and is continuous, the closed ball $B_\rho(\bar{a})$ with center $\bar{a}$ and radius $\rho$ is a compact set, and by the Bolzano–Weierstrass theorem [20], the Hessian norm must have a maximum value on this compact set, denoted as $L_H$. Therefore, the linearization error in the Taylor expansion of $Q(a)$ with respect to $a$ is bounded quadratically, satisfying the conditions for local linearity.

## H  Detailed Derivation of Intermediate Guidance

Implicit dependence of $Q(s, a_0)$ on action $a_0$ hinds the exact calculation of $\log \mathbb{E}_{\mu_{0|t}(a_0|a_t,s)}[e^{\beta Q(s,a_0)}]$. In order to approximate the intermediate energy $\mathcal{E}_t(s, a_t)$ shown in Equation (11), we first expand $Q(s, a_0)$ at $a_0 = \bar{a}$ with Taylor expansion, i.e.,

$$Q(s, a_0) \approx Q(s, a_0)|_{a_0=\bar{a}} + \frac{\partial Q(s, a_0)}{\partial a_0}|_{a_0=\bar{a}} * (a_0 - \bar{a}),$$

where $\bar{a}$ is a constant vector. Replacing it to $\log \mathbb{E}_{a_0 \sim \mu(a_0|a_t,s)} e^{\beta Q(s,a_0)}$, we can transfer the implicit dependence of $Q(s, a_0)$ on $a_0$ to explicit dependence on $Q'(s, \bar{a}) * a_0$. Accordingly,

$\log \mathbb{E}_{a_0 \sim \mu(a_0|a_t,s)} e^{\beta Q(s,a_0)}$ is given by

$$\log \mathbb{E}_{a_0 \sim \mu(a_0|a_t,s)} e^{\beta Q(s,a_0)}$$

$$\approx \log \mathbb{E}_{a_0 \sim \mu(a_0|a_t,s)} e^{\beta * \left( Q(s,a_0)|_{a_0=\bar{a}} + \frac{\partial Q(s,a_0)}{\partial a_0}|_{a_0=\bar{a}} * (a_0-\bar{a}) \right)}$$

$$= \log \mathbb{E}_{a_0 \sim \mu(a_0|a_t,s)} e^{\beta * \left( Q(s,\bar{a}) + Q'(s,\bar{a}) * (a_0-\bar{a}) \right)}$$

$$= \log \mathbb{E}_{a_0 \sim \mu(a_0|a_t,s)} \left[ e^{\beta * Q(s,\bar{a})} * e^{\beta * Q'(s,\bar{a}) * (a_0-\bar{a})} \right]$$

$$= \log \left\{ e^{\beta * Q(s,\bar{a}) - \beta * Q'(s,\bar{a}) * \bar{a}} \right\} + \log \left\{ \mathbb{E}_{a_0 \sim \mu(a_0|a_t,s)} [e^{\beta * Q'(s,\bar{a}) * a_0}] \right\}$$

$$= \beta * Q(s,\bar{a}) - \beta * Q'(s,\bar{a}) * \bar{a} + \log \left\{ \mathbb{E}_{a_0 \sim \mu(a_0|a_t,s)} [e^{\beta * Q'(s,\bar{a}) * a_0}] \right\}.$$

Furthermore, by applying the moment generating function of Gaussian distribution, we have

$$\log \mathbb{E}_{a_0 \sim \mu(a_0|a_t,s)} e^{\beta Q(s,a_0)}$$

$$= \beta * Q(s,\bar{a}) - \beta * Q'(s,\bar{a}) * \bar{a} + \log \left\{ \mathbb{E}_{a_0 \sim \mu(a_0|a_t,s)} [e^{\beta * Q'(s,\bar{a}) * a_0}] \right\}$$

$$= \beta * Q(s,\bar{a}) - \beta * Q'(s,\bar{a}) * \bar{a} + \beta * Q'(s,\bar{a})^\top * \tilde{\mu}_{0|t} + \frac{1}{2} * \beta^2 * \tilde{\sigma}_{0|t}^2 * ||Q'(s,\bar{a})||_2^2$$

$$= \beta * Q(s,\bar{a}) + \beta * Q'(s,\bar{a})^\top * (\tilde{\mu}_{0|t} - \bar{a}) + \frac{1}{2} * \beta^2 * \tilde{\sigma}_{0|t}^2 * ||Q'(s,\bar{a})||_2^2.$$

## I Detailed Derivation of Posterior Distribution

### I.1 Posterior 1

If we want to get the exact mean of $\mu_{0|t}(a_0|a_t,s)$, we actually want to optimize the following objective [6]

$$\tilde{\mu}_{0|t}^*(a_0|a_t,s) = \min_{\theta} \mathbb{E}_{a_0 \sim \mu_{0|t}(a_0|a_t,s)} \left[ ||\tilde{\mu}_{0|t,\theta}(a_0|a_t,s) - a_0||_2^2 \right], \tag{36}$$

where $\tilde{\mu}_{0|t,\theta}(a_0|a_t,s)$ can be further reparameterized by

$$\tilde{\mu}_{0|t,\theta} = \frac{1}{\alpha_t}(a_t - \sigma_t \epsilon_\theta(s, a_t, t)), \tag{37}$$

because this reparameterization follows the same loss function (as shown in Equation (7)) as the diffusion model, and we do not need to introduce additional parameters to approximate the mean of $\mu_{0|t}(a_0|a_t,s)$. For the covariance matrix $\tilde{\Sigma}_{0|t}$, following the definition of covariance

$$\tilde{\Sigma}_{0|t}(a_t) = \mathbb{E}_{\mu_{0|t}(a_0|a_t,s)} \left[ (a_0 - \tilde{\mu}_{0|t})(a_0 - \tilde{\mu}_{0|t})^\top \right]$$

$$= \mathbb{E}_{\mu_{0|t}(a_0|a_t,s)} \left[ (a_0 - \frac{1}{\alpha_t}(a_t - \sigma_t \epsilon_\theta))(a_0 - \frac{1}{\alpha_t}(a_t - \sigma_t \epsilon_\theta))^\top \right]$$

$$= \mathbb{E}_{\mu_{0|t}(a_0|a_t,s)} \left[ (a_0 - \frac{1}{\alpha_t}a_t)(a_0 - \frac{1}{\alpha_t}a_t)^\top \right] - \frac{\sigma_t^2}{\alpha_t^2} \epsilon_\theta \epsilon_\theta^\top \tag{38}$$

$$= \frac{1}{\alpha_t^2} \mathbb{E}_{\mu_{0|t}(a_0|a_t,s)} \left[ (a_t - \alpha_t a_0)(a_t - \alpha_t a_0)^\top \right] - \frac{\sigma_t^2}{\alpha_t^2} \epsilon_\theta \epsilon_\theta^\top,$$

where in the second equation we replace $\tilde{\mu}_{0|t}$ with Equation (37). In the third equation, we use

$$\mathbb{E}_{\mu_{0|t}(a_0|a_t)} [(a_0 - \frac{1}{\alpha_t}a_t) * \frac{\sigma_t}{\alpha_t} \epsilon_\theta]$$

$$= (\mathbb{E}_{\mu_{0|t}(a_0|a_t,s)} [a_0] - \frac{1}{\alpha_t}a_t) * \frac{\sigma_t}{\alpha_t} \epsilon_\theta$$

$$= (\tilde{\mu}_{0|t} - \frac{1}{\alpha_t}a_t) * \frac{\sigma_t}{\alpha_t} \epsilon_\theta$$

$$= -\frac{\sigma_t^2}{\alpha_t^2} \epsilon_\theta \epsilon_\theta^T,$$

and $\tilde{\mu}_{0|t} = \mathbb{E}_{\mu_{0|t}(a_0|a_t,s)}[a_0]$. Besides, we also notice that $\mu(a_t) \sim \mathcal{N}(a_t; \alpha_t a_0, \sigma_t^2 \boldsymbol{I})$, thus we have

$$\mathbb{E}_{\mu_t(a_t|s)}\mathbb{E}_{\mu_{0|t}(a_0|a_t,s)} \left[ (a_t - \alpha_t a_0)(a_t - \alpha_t a_0)^\top \right]$$
$$= \mathbb{E}_{\mu(a_0|s)}\mathbb{E}_{\mu_{t|0}(a_t|a_0,s)} \left[ (a_t - \alpha_t a_0)(a_t - \alpha_t a_0)^\top \right]$$
$$= \mathbb{E}_{\mu(a_0|s)}\mathbb{E}_{\mu_{t|0}(a_t|a_0,s)} \left[ (a_t - \alpha_t a_0)(a_t - \alpha_t a_0)^\top \right]$$
$$= \mathbb{E}_{\mu(a_0|s)}\Sigma_{t|0}$$
$$= \sigma_t^2 \boldsymbol{I},$$

where in the first equation $\mu_t(a_t|s)\mu_{0|t}(a_0|a_t,s) = \mu(a_0|s)\mu_{t|0}(a_t|a_0,s)$. To make the variance independent from the data $a_t$, we can apply the expectation over $\tilde{\Sigma}_{0|t}(a_t)$:

$$\tilde{\Sigma}_{0|t} = \mathbb{E}_{\mu_t(a_t|s)}\tilde{\Sigma}_{0|t}(a_t)$$
$$= \frac{\sigma_t^2}{\alpha_t^2}[\boldsymbol{I} - \mathbb{E}_{\mu_t(a_t|s)}[\epsilon_\theta(a_t, t)\epsilon_\theta(a_t, t)^\top]].$$

For simplicity, we usually consider isotropic Gaussian distribution, i.e., $\tilde{\Sigma}_{0|t} = \tilde{\sigma}_{0|t}^2 \boldsymbol{I}$, which indicates that

$$\tilde{\sigma}_{0|t}^2 = \frac{\sigma_t^2}{\alpha_t^2}[1 - \frac{1}{d}\mathbb{E}_{\mu_t(a_t|s)}[||\epsilon_\theta(a_t, t)||_2^2]].$$

## I.2 Posterior 2

For the mean value of the distribution $\mu_{0|t}(a_0|a_t, s)$, we still adopt the same reparameterization trick $\tilde{\mu}_{0|t,\theta} = \frac{1}{\alpha_t}(a_t - \sigma_t \epsilon_\theta(s, a_t, t))$. But for the covariance, we can reformulate [79] it as

$$\tilde{\Sigma}_{0|t}(a_t) = \mathbb{E}_{\mu_{0|t}(a_0|a_t,s)} \left[ (a_0 - \tilde{\mu}_{0|t})(a_0 - \tilde{\mu}_{0|t})^\top \right]$$
$$= \mathbb{E}_{\mu_{0|t}(a_0|a_t,s)} \left[ ((a_0 - u_0) - (\tilde{\mu}_{0|t} - u_0)) ((a_0 - u_0) - (\tilde{\mu}_{0|t} - u_0))^\top \right] \quad (39)$$
$$= \mathbb{E}_{\mu_{0|t}(a_0|a_t,s)} \left[ (a_0 - u_0)(a_0 - u_0)^\top \right] - (\tilde{\mu}_{0|t} - u_0)(\mu_{0|t} - u_0)^\top,$$

where we add a constant vector $u_0$ that has the same dimension with $\tilde{\mu}_{0|t}$ in the second equation. We use $\tilde{\mu}_{0|t} = \mathbb{E}_{\mu_{0|t}(a_0|a_t,s)}[a_0]$ to derivate the third equation. Perform expectation on $\tilde{\Sigma}_{0|t}(a_t)$ and let $u_0 = \mathbb{E}_{a_0 \sim \mu(a_0)}[a_0]$, we have

$$\tilde{\sigma}_t^2 = \mathbb{E}_{\mu_t(a_t|s)}[\tilde{\Sigma}_{0|t}(a_t)],$$
$$= \mathbb{E}_{\mu_t(a_t|s)}\mathbb{E}_{\mu_{0|t}(a_0|a_t,s)} \left[ (a_0 - u_0)(a_0 - u_0)^\top \right] - \mathbb{E}_{\mu_t(a_t|s)} \left[ (\tilde{\mu}_{0|t} - u_0)(\tilde{\mu}_{0|t} - \mu_0)^\top \right],$$
$$= \mathbb{E}_{\mu(a_0|s)}\mathbb{E}_{\mu_{t|0}(a_t|a_0,s)} \left[ (a_0 - u_0)(a_0 - u_0)^\top \right] - \mathbb{E}_{\mu_t(a_t|s)}(\tilde{\mu}_{0|t} - u_0)(\tilde{\mu}_{0|t} - u_0)^\top,$$
$$= \frac{1}{d}\mathbb{E}_{\mu(a_0|s)} \left[ ||a_0 - u_0||_2^2 \right] - \frac{1}{d}\mathbb{E}_{\mu_t(a_t|s)}[||\tilde{\mu}_{0|t} - u_0||_2^2],$$
$$= Var(a_0) - \frac{1}{d}\mathbb{E}_{\mu_t(a_t|s)}[||\tilde{\mu}_{0|t} - u_0||_2^2], \quad (40)$$

where $Var(a_0)$ can be approximated from a batch of data or from the entire dataset and $\mathbb{E}_{\mu_t(a_t|s)}[||\tilde{\mu}_{0|t} - u_0||_2^2]$ can be estimated by sampling a batch of data $a_0$ and $a_t$ with different $t$.

## I.3 Another Method to Approximate Posterior $\mu_{0|t}$

Consider the covariance estimation under $\mu_{0|t} \neq \mathbb{E}_{\mu_{0|t}(a_0|a_t,s)}[a_0]$ where $\mu_{0|t}$ has been trained well using the diffusion loss function (shown in Equation (7)), we can adopt the following negative log-likelihood optimization problem [5]

$$\min_{\tilde{\Sigma}_{0|t}} \mathbb{E}_{a_t \sim \mu_t(a_t|s), a_0 \sim \mu_{0|t}(a_0|a_t,s)}[- \log P(a_0; \tilde{\mu}_{0|t}, \tilde{\Sigma}_{0|t})].$$

perform derivation w.r.t $\tilde{\Sigma}_{0|t}$, we have

$$\frac{\partial}{\partial \tilde{\Sigma}_{0|t}} - \log \ P(a_0; \tilde{\mu}_{0|t}, \tilde{\Sigma}_{0|t}),$$

$$= \frac{\partial}{\partial \tilde{\Sigma}_{0|t}} - \log \left[ \frac{1}{2\pi^{d/2} |\tilde{\Sigma}_{0|t}|^{1/2}} e^{-\frac{1}{2}(a_0 - \tilde{\mu}_{0|t})^\top \tilde{\Sigma}_{0|t}^{-1}(a_0 - \tilde{\mu}_{0|t})} \right],$$

$$= \frac{\partial}{\partial \tilde{\Sigma}_{0|t}} \left[ \frac{d}{2} \log \ 2\pi + \frac{1}{2} \log |\tilde{\Sigma}_{0|t}| + \frac{1}{2}(a_0 - \tilde{\mu}_{0|t})^\top \tilde{\Sigma}_{0|t}^{-1}(a_0 - \tilde{\mu}_{0|t}) \right],$$

$$= \frac{1}{2}\tilde{\Sigma}_{0|t}^{-1} - \frac{1}{2}\tilde{\Sigma}_{0|t}^{-1}(a_0 - \tilde{\mu}_{0|t})(a_0 - \tilde{\mu}_{0|t})^\top \tilde{\Sigma}_{0|t}^{-1}.$$

So the objective reaches maximization when $\tilde{\Sigma}_{0|t}^* = \mathbb{E}_{a_t \sim \mu_t(a_t|s), a_0 \sim \mu_{0|t}(a_0|a_t,s)} \left[ (a_0 - \tilde{\mu}_{0|t})(a_0 - \tilde{\mu}_{0|t})^\top \right]$.
Usually, we consider the isotropic Gaussian distribution, i.e., $\tilde{\Sigma}_{0|t} = \tilde{\sigma}_{0|t}^2 I$ and

$$\tilde{\sigma}_{0|t}^2 = \frac{1}{d}\mathbb{E}_{a_0 \sim \mu(a_0|s), a_t \sim \mu_{t|0}(a_t|a_0,s)}[||a_0 - \tilde{\mu}_{0|t}||_2^2]. \tag{41}$$

Furthermore, we also have the following result, which is the same as Equation (41):

$$\tilde{\sigma}_{0|t}^2 = \frac{1}{d}\mathbb{E}_{a_0 \sim \mu(a_0|s), a_t \sim \mu_{t|0}(a_t|a_0,s)}[||a_0 - \tilde{\mu}_{0|t}||_2^2],$$

$$= \frac{1}{d}\mathbb{E}_{a_0 \sim \mu(a_0|s), \epsilon \sim \mathcal{N}(0,I), t \sim U[0,T], a_t = \alpha_t a_0 + \sigma_t \epsilon} \left[ ||a_0 - \frac{1}{\alpha_t}(a_t - \sigma_t \epsilon_\theta(s, a_t, t))||_2^2 \right], \tag{42}$$

$$= \frac{\sigma_t^2}{\alpha_t^2 d}\mathbb{E}_{a_0 \sim \mu(a_0|s), \epsilon \sim \mathcal{N}(0,I), t \sim U[0,T], a_t = \alpha_t a_0 + \sigma_t \epsilon} \left[ ||\epsilon - \epsilon_\theta(s, a_t, t)||_2^2 \right].$$

Inspired by the previous studies [5, 63] that $\tilde{\mu}_{0|t} = \mathbb{E}_{\mu_{0|t}(a_0|a_t,s)}[a_0]$ is suitable for training in practice, so we only focus on the performance of Posterior 1 and Posterior 2 in this paper.

We summarize the posterior mean and covariance in

$$\tilde{\mu}_{0|t} = \frac{1}{\alpha_t}(a_t - \sigma_t \epsilon_\theta(s, a_t, t)),$$

$$\tilde{\sigma}_{0|t}^2 = \frac{\sigma_t^2}{\alpha_t^2} \left[ 1 - \frac{1}{d}\mathbb{E}_{\mu_t(a_t|s)} \left[ ||\epsilon_\theta(a_t, t)||_2^2 \right] \right],$$

$$\tilde{\sigma}_{0|t}^2 = Var(a_0) - \frac{1}{d}\mathbb{E}_{\mu_t(a_t)}[||\tilde{\mu}_{0|t} - u_0||_2^2], \tag{43}$$

$$\tilde{\sigma}_{0|t}^2 = \frac{\sigma_t^2}{\alpha_t^2 d}\mathbb{E}_{a_0 \sim \mu(a_0|s), \epsilon \sim \mathcal{N}(0,I), t \sim U[0,T], a_t = \alpha_t a_0 + \sigma_t \epsilon} \left[ ||\epsilon - \epsilon_\theta(s, a_t, t)||_2^2 \right].$$

## J The Training of Q Function

### J.1 Q-function Training with Expectile Regression

Expectile regression loss

$$\mathcal{L}_V = \mathbb{E}_{(s,a) \sim D_\mu} \left[ L_2^\tau(V_\phi(s) - Q_{\bar{\psi}}(s, a)) \right],$$

$$\mathcal{L}_Q = \mathbb{E}_{(s,a,s') \sim D_\mu} \left[ ||r(s, a) + \gamma V_\phi(s') - Q_\psi(s, a)||_2^2 \right],$$

$$L_2^\tau(y) = |\tau - 1(y < 0)|y^2,$$

provides a method to avoid out-of-sample actions entirely and simultaneously perform maximization over the actions implicitly. The target value of Q is induced from a parameterized state value function, which is trained by expectile regression on in-sample actions. Besides, its effectiveness has been verified by many recent studies [53, 25]. Thus, we adopt this method to train the Q function.

## J.2 Q Function Training with In-support Softmax Q-Learning

Due to the natural property of data augmentation in generative models, such as diffusion models, we can use diffusion models to generate fake samples [64] and augment the training of the Q function:

$$\mathcal{L}_{ISQL} = \mathbb{E}_{(s,a,s') \sim D} \left[ ||Q_\psi(s,a) - \mathcal{T}^\pi Q_\psi(s,a)||_2^2 \right],$$

$$\mathcal{T}^\pi Q_\psi(s,a) \approx r(s,a) + \gamma * \frac{\sum_{\hat{a}'} \left[ e^{\beta Q_\psi(s',\hat{a}')} * Q_\psi(s',\hat{a}') \right]}{\sum_{\hat{a}'} e^{\beta Q_\psi(s',\hat{a}')}},$$

where $\hat{a}'$ is the fake actions. This method needs additional samples to train the Q function, where we can not make sure all the fake actions are reasonable for certain states. To a certain extent, these fake actions are OOD actions for the value functions. Besides, additional actions will also cause large memory consumption.

## J.3 Q Function Training with Contrastive Q-Learning

Contrastive Q-learning [55] is another notable method for training the Q function from offline datasets. It underestimates the action values for all actions that do not exist in the dataset to reduce the influence of OOD actions. The training loss is

$$\mathcal{L}_{CQL} = \mathbb{E}_{s,a,s' \sim D} \left[ ||Q_\psi(s,a) - (r + Q_{\bar{\psi}}(s', a' = \pi(a'|s')))||_2^2 \right]$$
$$+ \lambda \left( \mathbb{E}_{s \sim D, a \sim \pi(a|s)}[Q_\psi(s,a)] - \mathbb{E}_{(s,a) \sim D}[Q_\psi(s,a)] \right).$$

The learned Q function tends to predict similar values for actions that do not exist in datasets, which will cause ineffective intermediate guidance and further affect the performance.

# K Implementation Details

## K.1 Guidance Rescale Strategy

The guidance scale affects the amplitude and direction of the generation. The direction of generation must be changed due to the gradient guidance of the critic. So we can We find that when the guidance scale is zero, the inference performance is more stable than the guidance scale is non-zero, which inspires us to modify the amplitude of the score of the optimal data distribution $\nabla_{a_t} \log \pi_t(a_t|s)$. Mathematically, we re-normalize the amplitude of $\nabla_{a_t} \log \pi_t(a_t|s)$ by

$$\nabla_{a_t} \log \pi_t(a_t|s) = \frac{\nabla_{a_t} \log \pi_t(a_t|s)}{||\nabla_{a_t} \log \pi_t(a_t|s)||} * ||\nabla_{a_t} \log \mu_t(a_t|s)||. \tag{44}$$

## K.2 The Choice of $\bar{a}$ and $u_0$

Although we can choose $\bar{a}$ as any vector, considering the error bound of Taylor expansion and the stability of the training process, we use the following method to obtain $\bar{a}$

$$\bar{a} = a - \nu * \frac{Q'_\psi(s,a_0)}{||Q'_\psi(s,a_0)||}, \tag{45}$$

where $\nu = 0.001$. Considering the loss term of second-order approximation

$$R_2(a) = \frac{1}{2} \nabla_a^2 Q(s,c)(a - \bar{a})^2,$$

where $c$ is the point between $[\bar{a}, a]$. Considering the definition of Equation (45),

$$||\nabla_a Q(s,a)|| * ||a - \bar{a}||$$
$$= ||\nabla_a Q(s,a)|| * \nu * ||\frac{Q'_\psi(s,a_0)}{||Q'_\psi(s,a_0)||}||$$
$$= \nu.$$

So the error satisfies

$$R_2(a) = \frac{1}{2}\left[(\nabla_a Q(s,c))^T(a-\bar{a})\right]^2 \approx \frac{1}{2}\left[(\nabla_a Q(s,a))^T(a-\bar{a})\right]^2 \leq \frac{1}{2}\nu^2$$

as long as $\bar{a}$ is close to $a$, i.e., the derivative between $\bar{a}$ and $a$ is approximately constant. In practice, we use $Q'(s,a)$ as the value of $Q'(s,\bar{a})$ for two reasons: (1) Small $\nu$ value leads to approximately same derivative value at $a$ and $\bar{a}$. (2) Directly use $Q'(s,a)$ will reduce one calculation of gradient for $Q'(s,\bar{a})$.

In Equation (43), we approximate $\mathbb{E}_{q(a_t)}[||\epsilon_\theta(a_t,t)||_2^2]$ by sampling a batch of data and the calculation process is (1) Sample a batch of data $a_0$ from the dataset and sample $\epsilon$ from Gaussian distribution $\mathcal{N}(0,\boldsymbol{I})$. (2) Sample time step index $t$. (3) Obtain $a_t = \alpha_t a_0 + \sigma_t \epsilon$. (4) Get the noise prediction norm with diffusion model $\epsilon_\theta(a_t,t,s)$. (5) Average on the batch of data.

In Equation (43), $u_0$ serves as a constant baseline to calculate $\bar{\sigma}_t^2$, where $\mathbb{E}_{\mu(a_t)}[||\tilde{\mu}_{0|t}-u_0||_2^2]$ can be approximated from the data. (1) We can sample a batch size of data $a_0$ from the dataset and sample $\epsilon$ from Gaussian distribution $\mathcal{N}(0,\boldsymbol{I})$. (2) Sample time step index $t$. (3) Obtain $a_t = \alpha_t a_0 + \sigma_t \epsilon$. (4) Use the diffusion model to predict $\mu_{0|t} = \frac{1}{\alpha_t}(a_t - \sigma_t \epsilon_\theta(a_t,t,s))$. (5) Calculate $\mathbb{E}_{\mu(a_t)}[||\tilde{\mu}_{0|t}-u_0||_2^2]$ and average on the batch data.

