# OpenReview forum: "Analytic Energy-Guided Policy Optimization for Offline Reinforcement Learning"
_NeurIPS.cc/2025/Conference — NeurIPS 2025 poster_

### Official Review · Reviewer_eV4k · 2025-06-25

**Clarity:** 3
**Significance:** 3
**Originality:** 3
**Rating:** 4
**Confidence:** 3

**Summary:**

The paper introduces Analytic Energy-Guided Policy Optimization (AEPO), a novel method for offline reinforcement learning that leverages diffusion policy with analytically derived energy guidance. Inspired by the connection between guided diffusion sampling and constrained RL, the authors aim to improve action generation by analytically estimating the exact intermediate energy term, which is usually approximated or manually designed in prior work. By Taylor expansion of the Q value function and moment generating function of the Gaussian distribution, AEPO give an explicit estimation of the intermediate energy function during the diffusion sampling.

**Questions:**

1. The first two terms in Equation (15) assume that $\bar{a}$ and $a_0$ are independent. However, Appendix J.2 shows that the value of $\bar{a}$ depends on $a_0$. Does this affect the accuracy of Equation (15)?

2.  AEPO seems sensitive to several hyperparameters, such as the inverse temperature, expectile weight, and guidance degree. Does this sensitivity affect its usefulness across different tasks? How should these hyperparameters be chosen based on the characteristics of a task?

3. Does AEPO offer any computational efficiency improvements compared to other state-of-the-art methods?

**Ethical Concerns:**

["NO or VERY MINOR ethics concerns only"]

**Final Justification:**

Most of my concerns have been addressed by the author rebuttal!

**Limitations:**

Consider improving how the paper presents its motivation, especially the connection between guided diffusion sampling and constrained RL. Although Equation (1) and Equation (2) are similar, the paper should provide a clearer intuition for why the Q-value function can be used as an energy function.

**Paper Formatting Concerns:**

Double check the position of checklist.

**Quality:**

3

**Strengths And Weaknesses:**

### Strengths:

-	This paper derives a closed-form approximation of the intermediate energy used in guided diffusion policy sampling by Taylor expansion and moment-generating functions.
-	AEPO achieves better performance for many D4RL tasks compared with SOTA diffusion-based and non-diffusion-based offline RL methods.
-	This paper is well written and easy to follow, and it also provides detail proof about the theoretical results and experimental study.

### Weaknesses:

-	The closed-form approximation of the intermediate energy in AEPO heavily relies on the Taylor expansion of the Q-value function. However, the validity of a Taylor expansion requires more than differentiability—it also assumes that the first- and second-order terms dominate, and that higher-order terms are negligible. The paper lacks sufficient discussion or analysis on whether these conditions hold in practice. If the Taylor approximation deviates significantly from the true Q-value, then the analytically derived intermediate energy remains just an estimate. In that case, it may suffer from similar biases as previous approaches that relied on inexact intermediate energy approximations.

-	While AEPO introduces a theoretically grounded closed-form approximation of the intermediate energy, its empirical advantage over other SOTA diffusion policy methods is relatively modest. Notably, Appendix B.3 shows that the proposed Guidance Rescaling technique for the diffusion score function yields significant performance gains. This suggests that the reported improvements may be largely attributed to this auxiliary component, rather than the core contribution of the closed-form intermediate energy itself. As a result, the experimental comparisons may not be entirely fair or isolating the effect of the main innovation. Furthermore, the paper lacks ablation studies on the impact of different Q-function learning strategies, which would be important for understanding the robustness and general applicability of AEPO.

-	The paper omits several relevant and competitive baselines, particularly offline RL methods that incorporate uncertainty based pessimistic Q-value estimation, such as UWAC[1] and QDQ[2].

[1]  Wu, Y., Zhai, S., Srivastava, N., Susskind, J., Zhang, J., Salakhutdinov, R., & Goh, H. (2021). Uncertainty weighted actor-critic for offline reinforcement learning. arXiv preprint arXiv:2105.08140.

[2] Zhang, J., Fang, L., Shi, K., Wang, W., & Jing, B. (2024). Q-Distribution guided Q-learning for offline reinforcement learning: Uncertainty penalized Q-value via consistency model. Advances in Neural Information Processing Systems, 37, 54421-54462.

---

> ### Author Rebuttal · Authors · 2025-07-31
>
> Dear Reviewer eV4k,
>
> We appreciate your suggestions and respond to your concerns below.
>
> ### [I]. Explanation of Weaknesses
>
> > **[1/3] W1:** The closed-form approximation of the intermediate energy in AEPO heavily relies on the Taylor expansion of the Q-value function. However, the validity of a Taylor expansion requires more than differentiability—it also assumes that the first- and second-order terms dominate, and that higher-order terms are negligible. The paper lacks sufficient discussion or analysis on whether these conditions hold in practice. If the Taylor approximation deviates significantly from the true Q-value, then the analytically derived intermediate energy remains just an estimate. In that case, it may suffer from similar biases as previous approaches that relied on inexact intermediate energy approximations.
>
> We thank the reviewer for raising this important and technically insightful review. Through the choice of activation function and network structure, we can theoretically show that the Q-function is twice differentiable and smooth with respect to the action, which forms the basis for our local Taylor expansion (Equation (14)) [1]. In order to control the error bound of Taylor expansion, we use the following method to choose $\bar{a}$ and prove the error satisfies $R_2(a)\leq \frac{1}{2} \nu^2$. We would like to clarify that we have provided the detailed proof in Appendix J.2, but to make the current content clear and complete, we have rewritten the proof below. Additionally, we will emphasize this proof in the revised version.
>
> **Proof of Taylor expansion error control:**
>
> Considering the error bound of Taylor expansion and the stability of the training process, we use the following method to obtain $\bar{a}$:
> $\bar{a} = a - \nu*\frac{Q^\prime_{\psi}(s, a_0)}{||Q^\prime_{\psi}(s, a_0)||},$
> where $\nu=0.001$. Considering the loss term of the second-order approximation
> $R_2(a) = \frac{1}{2}\nabla_a^2 Q(s, c)(a-\bar{a})^2,$
> where $c$ is the point between $[\bar{a}, a]$. Thus,
> $||\nabla_a Q(s, a)||*||a-\bar{a}||=\nu,$
> So the error satisfies
> $R_2(a)\leq \frac{1}{2} \nu^2.$
>
> > **[2/3] W2:** While AEPO introduces a theoretically grounded closed-form approximation of the intermediate energy, its empirical advantage over other SOTA diffusion policy methods is relatively modest. Notably, Appendix B.3 shows that the proposed Guidance Rescaling technique for the diffusion score function yields significant performance gains. This suggests that the reported improvements may be largely attributed to this auxiliary component, rather than the core contribution of the closed-form intermediate energy itself. As a result, the experimental comparisons may not be entirely fair or isolating the effect of the main innovation. Furthermore, the paper lacks ablation studies on the impact of different Q-function learning strategies, which would be important for understanding the robustness and general applicability of AEPO.
>
> The purpose of using guidance rescale is to make our method less sensitive to $\omega$. The main contributions are our theoretical results of intermediate energy and systematic comparison results with dozens of baselines on 33 tasks.
>
> The results in Table 1 show that the best performance achieved via grid search under both guidance rescale and non-guidance rescale settings is comparable, indicating that guidance rescale is not the primary factor contributing to significant performance gains.
>
> **Table 1**: The best performance comparison of guidance rescale and non-guidance rescale with grid search.
> |Type|guidance rescale|non-guidance rescale|
> |-|-|-|
> |walker2d-me|109.3±0.5|109.4±0.9|
> |walker2d-mr|90.8±1.5|92.8±4.8|
> |halfcheetah-mr|43.7±1.3|43.3±1.5|
>
> We want to clarify that regardless of which Q function training method is used, as long as the Q function learning is suitable for offline RL, the performance obtained using our method is similar. For example, with IQL-like Q function training methods and CQL-like Q function training methods, when obtaining accurately estimated Q functions, the performance of AEPO is similar. As shown by the experimental results in Table 2, this experiment used two methods to train the Q function, one being an IQL-like training method and the other using a CQL-like training method. We find that the experimental performance was similar, indicating that we just need to adopt the Q-function training method that is suitable and widely used in offline RL.
>
> **Table 2**: The influence of different training methods on the Q function. The experiments are conducted on AEPO but with different training methods of Q functions.
> |Q function training type|IQL-like Q function training|CQL-like Q function training|
> |-|-|-|
> |walker2d-me|109.3±0.5|109.8±0.7|
> |halfcheetah-me|94.4±0.9|93.8±0.9|
> |hopper-me|111.5±1.1|111.2±2.4|
>
> > **[3/3] W3:** The paper omits several relevant and competitive baselines, particularly offline RL methods that incorporate uncertainty based pessimistic Q-value estimation, such as UWAC[2] and QDQ[3].
>
> We thank the reviewer for the valuable advice.
> We would like to clarify that the research focus of our paper is not on Q-value estimation, but rather on the diffusion model's guided-sampling with intermediate energy. Both of these reference papers focus on Q-value estimation. Our research emphasis is distinctly different.
>
> Following your suggestion, we will add experimental comparisons with UWAC and QDQ in the main body. Table 3 illustrates a significant improvement in Locomotion tasks by comparing our method with QDQ. From Table 4, we can see that our method surpasses UWAC on 10/12 Adroit tasks and reaches a higher mean score than UWAC.
>
> **Table 3**: The experiments comparison on the Locomotion tasks.
> |Model|Antmaze-umaze|Antmaze-umaze-diverse|Antmaze-medium-play|Antmaze-medium-diverse|Antmaze-large-play|Antmaze-large-diverse|mean score|
> |-|-|-|-|-|-|-|-|
> |QDQ|98.6|67.8|81.5|85.4|35.6|31.2|66.7|
> |AEPO|100.0|100.0|76.7|83.3|56.7|66.7|80.6|
>
> **Table 4**: The experiments comparison on the Adroit tasks.
> |Model|pen-h|pen-e|pen-c|hammer-h|hammer-e|hammer-c|door-h|door-e|door-c|relocate-h|relocate-e|relocate-c|mean score|
> |-|-|-|-|-|-|-|-|-|-|-|-|-|-|
> |UWAC|65.0|119.8|45.1|8.3|128.8|1.2|10.7|105.4|1.2|0.5|108.7|0.0|49.6|
> |AEPO|76.7|147.0|69.3|10.3|129.7|6.4|9.0|106.5|3.9|0.8|107.0|0.6|55.6|
>
> ### [II]. Explanation of Questions
>
> > **[1/3] Q1:** The first two terms in Equation (15) assume that
>  $\bar{a}$ and $a_0$ are independent. However, Appendix J.2 shows that the value of $\bar{a}$ depends on $a_0$. Does this affect the accuracy of Equation (15)?
>
> We want to clarify that in Equation (15), we did not assume that $\bar{a}$ and $a_0$ are independent. In Equation (15), $a_0$ is a general notation symbol, while $\bar{a}$ represents any constant vector. In Appendix J.2, to control the loss term of second-order approximation, we introduce $\nu$ and use $a_0$ to calculate $\bar{a}$.
>
> > **[2/3] Q2:** AEPO seems sensitive to several hyperparameters, such as the inverse temperature, expectile weight, and guidance degree. Does this sensitivity affect its usefulness across different tasks? How should these hyperparameters be chosen based on the characteristics of a task?
>
> In the experiments, the values of inverse temperature $\beta$ and expectile weight $\tau$ are $\beta$=1 and $\tau$=0.5, which is suitable for most tasks. Regarding guidance degree $\omega$, to make our method robust to $\omega$ values, we propose guidance rescale, and experimentally demonstrate that guidance rescale can make our method robust for $\omega$ in a wider range. Usually, $\omega=0.1$ is appropriate for most tasks after guidance rescale.
>
> > **[3/3] Q3:** Does AEPO offer any computational efficiency improvements compared to other state-of-the-art methods?
>
> Although we acknowledge that AEPO does not offer explicit computational efficiency improvements over existing SOTA methods, we would like to emphasize that we provide **rigorous theoretical results regarding intermediate energy**, and have compared with **30+ baselines** on **33 tasks**, achieving SOTA performance on most tasks, demonstrating the enormous potential of our method.
>
> ### [III]. Explanation of Limitations
> > **[1/1] L1:** why the Q-value function can be used as an energy function.
>
> Thank you for your insightful comments. We try to address your concern by clarifying the intuition behind using the Q-value function as an energy function.
>
> In reinforcement learning, the Q-value Q(s,a) represents the expected return of taking action a in state s. Higher Q-values indicate more desirable actions. In energy-based guided sampling, the energy function E(x) serves as a scoring function, where lower energy corresponds to higher sampling probability on x.
> By identifying the energy function as E(x)=−Q(s,a), we make high-Q actions more likely to be sampled through the exponential weighting $\beta$Q(s,a). This aligns with the optimization objective in constrained RL: maximizing the expected Q-value under a KL divergence constraint from a prior behavior policy.
> We will clarify this relation and the underlying intuition more explicitly in the revised version to help readers better understand the connection between energy-function-guided diffusion models and the constrained RL problem.
>
> ### [IV]. Explanation of Paper Formatting
>
> Thank you for your careful review. We will place the checklist behind the appendix in the revised version.
>
> **References**
>
> [1] Principles of mathematical analysis.
>
> [2] Uncertainty weighted actor-critic for offline reinforcement learning.
>
> [3] Q-Distribution guided Q-learning for offline reinforcement learning: Uncertainty penalized Q-value via consistency model.

---

> > ### Comment · Reviewer_eV4k · 2025-08-05
> >
> > Thank you for the detailed reply, most of my concerns have been addressed. I will raise my score!

---

> > > ### Author Response · Authors · 2025-08-05
> > > **Thanks to Reviewer eV4k**
> > >
> > > Dear Reviewer eV4k,
> > >
> > > Thank you very much for your kind and encouraging response.
> > >
> > > We are grateful that most of your concerns have been addressed and sincerely appreciate your decision to raise the score. Your valuable suggestions and positive feedback have helped us significantly improve the paper's quality.
> > >
> > > Thanks again for your time and support.
> > >
> > > Kind regards,
> > >
> > > Paper6763 Authors

---

### Official Review · Reviewer_THAc · 2025-06-30

**Clarity:** 2
**Significance:** 2
**Originality:** 4
**Rating:** 4
**Confidence:** 4

**Summary:**

This paper proposes an posterior estimation method for the exact energy guidance function if Diffusion RL. There are three key estimation employed in this paper. 1) Consider Q function as a linear function within a certain area. 2) Consider diffusion posterior \mu(\x_0| \x_t) as a Gaussian distribution. With these two estimations, the method can effective compute exact energy function during training, and apply energy guidance in Offline RL tasks such as D4RL.

**Questions:**

1. In Eq 3, Do you mean s.t. KL(\pi|\mu) <=  \eta?   It seems  <= xxx is missing.
2. Why cannot you analytically compute the gradient of energy function from Eq. 17.
3. Besides performance, what is the advantage of AEPO compared with previous work , especially QGPO and DPS.

**Ethical Concerns:**

["NO or VERY MINOR ethics concerns only"]

**Limitations:**

yes

**Quality:**

3

**Strengths And Weaknesses:**

Strengths:
1. The paper poses a neat and in-depth analysis of existing energy guidance method. The proposed method is well theoretical-grounded. The posterior estimation method contain good mathematical insight.
2. The experiments are extensive and supportive. This work compares with a series of related baselines to prove the effectiveness of the proposed method.

Weaknesses:
1) I think there's still room for improvement on the paper writing. E.g., What is the core motivation for proposing AEPO, what problem of QGPO(CEP) is AEPO trying to solve? This one should be highlighted in Introduction and Method Section.  Also, what is the key difference between DPS and AEPO (I think they both some kind of posterior estimation). Finally I would suggest keeping only \pi, \mu notation or p, q notation in introduction. Keeping both is not really necessary.
2) Regarding (1).  I think usually the advantage of Analytic xxx Algorithm is that you can compute some quantity so that you do not need to estimate them using a network. However, AEPO seems also need to train an energy guidance model (Eq. 25), which somehow weakens its advantage.
3) Line 177:   "Reminding that the posterior distribution µ0|t(a0|at, s) is also a Gaussian distribution". This is wrong.  $µ0|t(a0|at, s)$ is not Gaussian, $µ0|t(am|at, s)$ can be considered as Gaussian only if m and t are close.

---

> ### Author Rebuttal · Authors · 2025-07-31
>
> Dear Reviewer THAc,
>
> We appreciate your suggestions and respond to your concerns below.
>
> ### [I]. Explanation of Weaknesses
>
> > **[1.1/3] W1.1:** What is the core motivation for proposing AEPO, what problem of QGPO(CEP) is AEPO trying to solve? This one should be highlighted in Introduction and Method Section.
>
> Thank you for your suggestions.
>
> The core motivation of our method is that QGPO cannot solve the intractable intermediate energy with the log-expectation form, which leads to sub-optimal performance. Facing this issue, we resolve the intractable log-expectation of intermediate energy with Taylor expansion and the moment generating function, achieving better performance in 33 tasks with various domains.
>
> We discuss the differences between our method and QGPO in detail, and we will add the contents of core motivation in the Introduction and Method.
> - **From the perspective of theory,** our method provides a theoretical solution for the log-expectation of the intermediate energy, which takes a further step compared with QGPO. QGPO only derives the formula of intermediate energy rather than the further results of the log-expectation.
> - **From the perspective of algorithmic design,** in Theorem 3.2 of QGPO, where the authors use contrastive learning loss to fit the intermediate energy, the authors mention that, under infinite data and model capacity, the contrastive loss (Equation (12) in QGPO) can perfectly approximate the intermediate energy. However, in practice, we cannot achieve infinite model capacity, and offline samples are certainly limited. Our method does not require assumptions about the sample size or model capacity.
> - **From the perspective of dataset,** QGPO requires the use of a diffusion model to generate fake action vectors for CEP, and these fake action vectors may introduce the influence of out-of-distribution actions. However, our method does not require the generation of fake actions.
>
> > **[1.2/3] W1.2:** What is the key difference between DPS and AEPO (I think they both some kind of posterior estimation).
>
> The key difference between DPS and AEPO is: The intermediate energy formula used in AEPO is exact while that in DPS is inexact. The derivation of the exact and inexact intermediate energy can be found in Table 1 and Appendix E of QGPO. AEPO first bases on the exact intermediate energy, then solves the log-expectation that QGPO does not resolve.
>
> > **[1.3/3] W1.3:** Finally I would suggest keeping only \pi, \mu notation or p, q notation in introduction. Keeping both is not really necessary.
>
> Simultaneously introducing $\pi$, $\mu$ and p, q is intended to help readers intuitively see the direct relationship between them from a formal perspective, which can reduce, to some extent, the difficulty of understanding for readers. Following your suggestion, we will consider introducing $\pi$, $\mu$ in the introduction considering that this paper belongs to RL field, and then placing the relationship between $\pi$, $\mu$ and p, q in the preliminary section.
>
>
> > **[2/3] W2:** Regarding (1). I think usually the advantage of Analytic xxx Algorithm is that you can compute some quantity so that you do not need to estimate them using a network. However, AEPO seems also need to train an energy guidance model (Eq. 25), which somehow weakens its advantage.
>
> The reason we use "Analytic" is that our intermediate energy is derived from a theoretically grounded, analytically tractable formulation based on the Taylor approximation and the moment-generating function. This is in contrast to previous methods that rely on contrastive objectives such as CEP.
>
> To avoid potential ambiguity in the title, we accept your suggestions to remove 'Analytic', resulting in the new title 'Energy-Guided Policy Optimization for Offline Reinforcement Learning'.
>
>
> > **[3/3] W3:** Line 177: "Reminding that the posterior distribution µ0|t(a0|at, s) is also a Gaussian distribution". This is wrong. µ_{0|t}(a_0|a_t,s) is not Gaussian, µ_{0|t}(a_m|a_t,s) can be considered as Gaussian only if m and t are close.
>
> We thank the reviewer for the thorough and careful review. We will modify this sentence to: "Following previous research [1,2], we use a Gaussian distribution as the estimate for the posterior distribution $μ_{0|t}(a_0|a_t,s)$."
>
> ### [II]. Explanation of Questions
>
> > **[1/3] Q1:** In Eq 3, Do you mean s.t. KL(\pi|\mu) <= \eta? It seems <= xxx is missing.
>
> Yes, you are right. We will revise the equation as $D_{KL}(\pi|\mu) \leq \eta$.
>
> > **[2/3] Q2:** Why cannot you analytically compute the gradient of energy function from Eq. 17.
>
> We would like to clarify that Equation (17) is used to calculate the posterior mean vector of the intermediate energy rather than the gradient of the energy function. We explain the reasons as follows.
> 1) From Equation (11), the difficulty in differentiating the intermediate energy lies in the log-expectation, rather than $a_0$.
> 2) Equation (17) is mainly used to estimate the mean of the posterior distribution, while actually generating a_0 requires multiple steps; if we directly backpropagate through $a_0$, the computational graph would consume significant computational overhead due to the multi-step generation process.
>
> > **[3/3] Q3:** Besides performance, what is the advantage of AEPO compared with previous work , especially QGPO and DPS.
>
> We provide further discussion about the differences between AEPO, QGPO, and DPS in detail.
> - **From the intermediate energy guidance perspective**, AEPO and QGPO both use the exact intermediate energy, while DPS uses the inexact intermediate energy.
> - **From the theoretical perspective**, our method provides a theoretical solution for the log-expectation of the intermediate energy, which takes a further step compared with QGPO. QGPO only derives the formula of intermediate energy rather than the further results of the log-expectation.
> - **From the implementation perspective**: In Theorem 3.2 of QGPO, where the authors use contrastive learning loss to fit the intermediate energy, the authors mention that, under infinite data and model capacity, the contrastive loss (Equation (12) in QGPO) can perfectly approximate the intermediate energy. However, in practice, we cannot achieve infinite model capacity, and offline samples are certainly limited. Our method does not require assumptions about the sample size or model capacity.
> - **From the dataset perspective**: Furthermore, QGPO requires the use of a diffusion model to generate fake action vectors for CEP, and these fake action vectors may introduce the influence of out-of-distribution actions. However, our method does not require the generation of fake actions.
>
> **References**
>
> [1] Score-Based Generative Modeling through Stochastic Differential Equations.
>
> [2] Analytic-DPM: an Analytic Estimate of the Optimal Reverse Variance in Diffusion Probabilistic Models.

---

> > ### Comment · Reviewer_THAc · 2025-08-07
> >
> > Thank you for the detailed rebuttal. I'm happy with the comment.
> > Please make sure to revise the factual error mentioned in the previous review. This can be very misleading for those who are not entirely familiar with theoretical derivations.

---

> > > ### Author Response · Authors · 2025-08-07
> > > **Thanks to Reviewer THAc**
> > >
> > > Dear Reviewer THAc,
> > >
> > > Thank you very much for your constructive suggestions, which will greatly improve the quality and clarity of our paper. We will revise all related contents in the manuscript according to your suggestions, especially the theory-related contents, such as the issues of introduction, the title, and the contents of posterior distribution you suggested previously.
> > >
> > > Thanks again for your time and support!
> > >
> > > Kind regards,
> > >
> > > Paper6763 Authors

---

### Official Review · Reviewer_z6K5 · 2025-07-02

**Clarity:** 4
**Significance:** 3
**Originality:** 3
**Rating:** 5
**Confidence:** 2

**Summary:**

This paper proposes Analytic Energy-guided Policy Optimization (AEPO), a method designed to approximate the intermediate energy used in conditional decision generation with diffusion models. It first analyzes the limitations of existing methods that provide only inexact intermediate energy estimates, and then introduces a novel approximation technique based on Taylor expansions. Subsequently, the model approximates the Gaussian mean and covariance of the posterior distribution.
AEPO is evaluated on the D4RL offline reinforcement learning benchmark and shows promising performance compared to 30 baseline approaches.

**Questions:**

Except for D4RL, are there any other benchmarks that can be used to evaluate the current approach? Is the OG-Bench (ICLR25), a goal-conditioned offline RL benchmark, a reasonable choice for the current approach?

**Ethical Concerns:**

["NO or VERY MINOR ethics concerns only"]

**Final Justification:**

I recommend this paper due to its overall quality, I remain my 5 score as it is.

**Limitations:**

yes, a limitation section has been provided in the appendix.

**Paper Formatting Concerns:**

No concern.

**Quality:**

4

**Strengths And Weaknesses:**

# Strengths
- The paper presents comprehensive mathematical derivations, including key theorems, proofs, and approximations. I reviewed most of the proofs and found them technically solid and convincing, demonstrating a high level of quality.
- The experimental section is thorough, with comparisons against more than 30 baseline approaches on the D4RL benchmark, highlighting the competitiveness of the proposed method.
- A reasonable ablation study and analysis are provided to support the understanding of individual components.

# Weakness
- No major weaknesses are identified at this stage, acknowledging the reviewer's limited expertise in energy-based methods and offline RL. However, I remain open to further discussion with other reviewers regarding potential limitations of the paper during the rebuttal phase.

---

> ### Author Rebuttal · Authors · 2025-07-31
>
> Dear Reviewer z6K5,
>
> We appreciate your suggestions and respond to your concerns below.
>
>
> ### [I]. Explanation of Questions
>
> > **[1/1] Q1:** Except for D4RL, are there any other benchmarks that can be used to evaluate the current approach? Is the OG-Bench (ICLR25), a goal-conditioned offline RL benchmark, a reasonable choice for the current approach?
>
> We select D4RL to evaluate our method because it is widely adopted as a general-purpose benchmark for offline RL and widely used in subsequent literature [1]. D4RL encompasses a diverse suite of tasks across domains (e.g., locomotion, manipulation, maze navigation) and contains varying data quality levels (e.g., random, medium, expert, medium-replay), making it well-suited to examine the effectiveness of offline RL algorithms.
>
> We are aware that OG-Bench is a recently proposed benchmark specifically designed for goal-conditioned offline RL, which is an important and emerging subdomain of offline RL [2]. OG-Bench is more specialized and does not provide a general-purpose evaluation suite like D4RL. We evaluate our method on 33 tasks across different D4RL domains, covering a broad range of difficulties and data distributions. We believe this provides a strong empirical basis for evaluating the effectiveness of our approach.
>
> **References**
>
> [1] D4rl: Datasets for deep data-driven reinforcement learning.
>
> [2] Ogbench: Benchmarking offline goal-conditioned rl.

---

> > ### Comment · Reviewer_z6K5 · 2025-08-04
> >
> > Hi,
> >
> > Thank you for your reply, I will keep my positive recommendation of this paper, due to the facts that:
> > - The paper is in overall good quality.
> > - I am aware that there are not many options in choosing offline RL benchmarks. D4RL is unfortunately still the most widely used one.
> > - I read the other reviews and rebuttals and do not find very critical issues.

---

> > > ### Author Response · Authors · 2025-08-05
> > > **Thanks to Reviewer z6K5**
> > >
> > > Dear Reviewer z6K5,
> > >
> > > We are truly grateful for your positive recommendation and your recognition of the overall quality of our work. We are grateful for your time and effort throughout the review process.
> > >
> > > Thanks again for your thoughtful and supportive feedback!
> > >
> > > Kind regards,
> > >
> > > Paper6763 Authors

---

### Official Review · Reviewer_kfG8 · 2025-07-08

**Clarity:** 4
**Significance:** 4
**Originality:** 4
**Rating:** 5
**Confidence:** 4

**Summary:**

In this paper, the authors study the problem of offline RL with diffusion-model-based algorithms. It falls in the category of learning a diffusion model for the *behaviour policy*, using an energy function based on the Q-function to guide the generation. The paper identifies that the theoretically optimal guidance signal for this task, termed the *intermediate energy*, is defined by an intractable log-expectation formulation. Consequently, prior methods have relied on theoretically inexact heuristics or potentially unstable training procedures to approximate this guidance, creating a gap between theory and practice.

The proposed algorithm, Analytic Energy-guided Policy Optimization (AEPO), is novel with tractable analytic approximation for this intermediate energy. The authors achieve this by linearizing the Q-function with a first-order Taylor expansion and then leveraging the closed-form moment-generating function of a Gaussian distribution to resolve the expectation. Experiments have been conducted to validate the idea.

**Questions:**

1. How is the locality and smoothness of the Q-function (Equation 14) validated, both theoretically and experimentally?
2. Can the proposed AEPO be naturally extended to a discrete action space?

**Ethical Concerns:**

["NO or VERY MINOR ethics concerns only"]

**Final Justification:**

All concerns have been addressed during discussion and no change on reviews.

**Limitations:**

yes

**Paper Formatting Concerns:**

No formatting concerns

**Quality:**

3

**Strengths And Weaknesses:**

### Strengths

* This paper studies a highly significant and fundamental problem in diffusion-based reinforcement learning: the intractability of the theoretically optimal guidance signal.
* They propose a principled and effective solution. The core contribution—deriving an analytic approximation for the intermediate energy—is highly original and technically sound.
* The use of a Taylor expansion combined with the moment-generating function of a Gaussian is a clever and novel approach that elegantly transforms an intractable problem into a tractable one. It is a distinct and well-reasoned improvement over previous heuristics (like MSE/DPS) and more complex training schemes (like contrastive learning), and its derivation is clear and well-supported.
*  The paper is very clearly written and well-organized.

### Weaknesses

The paper presents several areas for improvement.

First, the first-order Taylor expansion of the Q-function in Equation (14) assumes local linearity. The authors should clarify whether the necessary conditions of Q-function smoothness and the locality of $\bar{a}$ around $a_0$ are met.

Second, the method's reliance on Gaussian mechanics restricts its applicability to continuous action spaces. This significant limitation is not adequately discussed in the main paper. While it may perform well in continuous spaces, its effectiveness in discrete spaces remains unclear.

Third, regarding experiments, while the results are generally strong, further analysis and experiments on computational efficiency are needed. Given the potentially long training times, more profiling would help identify areas for improvement.

Finally, the evaluation lacks diversity, focusing on a specific experimental setting rather than including real-world robotic tasks or domains with different physical properties. This narrow scope limits the assessment of the method's broader generalization and robustness, thereby reducing the stated impact of the results.

---

> ### Author Rebuttal · Authors · 2025-07-31
>
> Dear Reviewer kfG8,
>
> We appreciate your suggestions and respond to your concerns below.
>
> ### [I]. Explanation of Weaknesses
>
> > **[1/4] W1:** First, the first-order Taylor expansion of the Q-function in Equation (14) assumes local linearity. The authors should clarify whether the necessary conditions of Q-function smoothness and the locality of $\bar{a}$ around $a_0$ are met.
>
> Thanks for the rigorous comments.
> **For smoothness:** We want to clarify that we often use neural networks to learn the Q function in practical implementation, and by using specific activation functions, such as SiLU (the activation function used in our source code), we can ensure that the Q function is smooth with respect to actions and at least twice differentiable. This is because: If the activation function $\phi(\cdot)$ is SiLU, then $\phi(\cdot)$ is infinitely differentiable with respect to the input. Assuming the neural network is an L-layer feedforward network, i.e., $Q(a)=W_L\phi(W_{L-1}\phi(…..))$, then 1) affine transformations are infinitely differentiable, 2) activation functions are infinitely differentiable, so the composite function $Q(a)$ with respect to $a$ is also infinitely differentiable, and at least twice differentiable. Even in cases where activation functions like ReLU are used, the Q-function remains piecewise linear and locally approximable by linear functions in regions away from non-differentiable points.
>
> **For local linearity:** Since the Hessian of Q(a) with respect to a exists and is continuous, the closed ball $B_\rho(\bar{a})$ with center $\bar{a}$ and radius $\rho$ is a compact set, and by the Bolzano–Weierstrass theorem [1], the Hessian norm must have a maximum value on this compact set, denoted as $L_H$. Therefore, the linearization error in the Taylor expansion of $Q(a)$ with respect to $a$ is bounded quadratically, satisfying the conditions for local linearity.
>
> We will add the above discussion in the revised version.
>
>
> > **[2/4] W2:** Second, the method's reliance on Gaussian mechanics restricts its applicability to continuous action spaces. This significant limitation is not adequately discussed in the main paper. While it may perform well in continuous spaces, its effectiveness in discrete spaces remains unclear.
>
> Thanks for your insightful comments. We agree that the current method is primarily designed for continuous action spaces, where the use of the first-order Taylor expansion of the Q-function with respect to actions is mathematically justified due to the differentiability of Q(s, a) with respect to the continuous action variable.
>
> In discrete action spaces, however, there are several key challenges:
> - The use of diffusion models in discrete domains remains an open and actively studied research area. Modeling and sampling in discrete diffusion processes (e.g., categorical diffusion) are non-trivial and require additional mechanisms;
> - More fundamentally, the Q-function is not differentiable with respect to discrete actions, making the Taylor expansion mathematically difficult to interpret.
>
> We believe that it is an interesting and important future direction to extend our approach to discrete domains. We will explicitly acknowledge and discuss this limitation in the revised version and clarify that our current method is intended for environments with continuous action spaces.
>
> > **[3/4] W3:** Third, regarding experiments, while the results are generally strong, further analysis and experiments on computational efficiency are needed. Given the potentially long training times, more profiling would help identify areas for improvement.
>
> We select hopper-m as the test environment, set the batch size to 32, and control the same hardware conditions. The GPU is NVIDIA A10, and the CPU is Intel(R) Xeon(R) Gold 6230 CPU @ 2.10GHz.
> The results are shown in Table 1, where we report the results of runtime GPU memory usage, inference time consumption of every decision, and training time consumption of every neural network update. The results indicate that our method incurs lower GPU memory overhead compared to QGPO, primarily because it does not require loading additional 'fake_actions' data.
> In terms of training time, our method introduces higher overhead than QGPO, mainly due to the extra computation involved in calculating the intermediate energy of Equation (16). However, in inference time, the cost of AEPO and QGPO is comparable, as both adopt the same number of generation steps.
>
>
> **Table 1:** The computational efficiency comparison between AEPO and the SOTA diffusion-based model QGPO. We set the batch size to 32 and conduct experiments of hopper-m under the same hardware conditions. The GPU used is NVIDIA A10, and the CPU is Intel(R) Xeon(R) Gold 6230 CPU @ 2.10GHz.
> |Method|AEPO|QGPO|
> |-|-|-|
> |training time of every updating on hopper-m (s)|0.018|0.015|
> |GPU memory usage with batch size 32 on hopper-m (M)|1135|1161|
> |inference time of every generation on hopper-m (s)|0.068|0.070|
>
> > **[4/4] W4:** Finally, the evaluation lacks diversity, focusing on a specific experimental setting rather than including real-world robotic tasks or domains with different physical properties. This narrow scope limits the assessment of the method's broader generalization and robustness, thereby reducing the stated impact of the results.
>
> Thank you for your thoughtful and constructive feedback. We agree that the current evaluation is limited to virtual robotic environments, and thus may not fully reflect the method's potential generalization to real-world robotic systems or environments with diverse dynamics. Our primary motivation was to first validate the effectiveness of our method in standardized and widely-used simulation environments, which is beneficial to compare our method with previous representative methods.
> Extending the method to real-world robots or environments with significantly different physical properties is indeed an important and meaningful direction. However, we acknowledge that such an extension may involve several non-trivial challenges, including but not limited to:
> - Hardware and resource constraints: Real-world experiments require access to diverse robotic hardware, which can be costly to acquire and maintain. Additionally, the time, personnel, and safety considerations involved in such deployments often pose significant barriers.
> - Safety and deployability: Sampling-based or diffusion-driven policy outputs need to be carefully constrained to ensure stable and responsive behavior in real-world deployments.
> - Sim-to-real transferring: Offline datasets collected in one environment may not generalize well to novel states or transitions in real scenarios;
>
> We are happy to explore these directions in future work and evaluate our method under more diverse and challenging physical conditions, including real-robot settings.
>
>
> ### [II]. Explanation of Questions
>
> > **[1/2] Q1:** How is the locality and smoothness of the Q-function (Equation 14) validated, both theoretically and experimentally?
>
> We appreciate the reviewer’s insightful review. We would like to clarify that the locality and smoothness of the Q-function are theoretical conditions required to justify the Taylor expansion used in the derivation of the intermediate energy. These are not properties that can be directly validated or falsified through empirical experiments due to the following reasons:
> - Locality and smoothness are inherently infinitesimal properties. In contrast, experiments are based on finite samples and cannot assess functional behavior in an arbitrarily small neighborhood.
> - Numerical estimation of gradients or higher-order derivatives in high-dimensional action spaces is noisy, unstable, and lacks reproducibility across different runs or data subsets.
>
> From a practical standpoint, the smoothness of the Q-function is an inductive bias implicitly introduced through the architecture of the neural network. In practice, the Q-function is parameterized by a deep neural network with smooth activation functions, such as SiLU, which ensures differentiability and enables local linear approximations theoretically (Kindly refer to W1 for the theoretical analysis).
>
> > **[2/2] Q2:** Can the proposed AEPO be naturally extended to a discrete action space?
>
> If we want to apply our method to discrete action spaces, we believe the following research steps should be followed:
> 1) First, the guidance formula in equation (8) has a universal formula and can be used for both continuous action spaces and discrete action spaces.
> 2) When using diffusion models in disperse action spaces, the forward process is no longer in the continuous form as in equation (5), but instead uses random replacement to substitute the input x.
> 3) In this case, the intermediate energy formula in RL will no longer need to transform the log-expectation into a simplified form through Taylor expansion of the Q function and moment generating function, but instead approximates the log-expectation through sampling and averaging, because actions are discrete, and integration is directly replaced by summation.
> 4) Use neural networks to fit the simplified intermediate energy, just as in equation (25).
> 5) Implement guided generation using the intermediate energy fitted by neural networks.
>
>
> **References**
>
> [1] Principles of mathematical analysis.

---

> > ### Comment · Reviewer_kfG8 · 2025-08-07
> >
> > Thanks for authors responses. All my concerns here are addressed. As I'm inclined to accept this work, I'll keep my score and reviews

---

> > > ### Author Response · Authors · 2025-08-07
> > > **Thanks to Reviewer kfG8**
> > >
> > > Dear Reviewer kfG8,
> > >
> > > We are grateful that your concerns have been addressed, and we sincerely appreciate your constructive feedback throughout the review process. We will carefully revise the manuscript to incorporate the corresponding suggestions provided in your review.
> > >
> > > Thanks again for your time and thoughtful comments!
> > >
> > > Kind regards,
> > >
> > > Paper6763 Authors

---

### Decision · Program_Chairs · 2025-09-17

**Decision:**

Accept (poster)

**Comment:**

This paper studies the problem of offline RL with diffusion-model-based algorithms, where a diffusion model is learned for the behavior policy and guided by an energy function derived from the Q-function. A key challenge is that the theoretically optimal guidance signal—the intermediate energy—is defined by an intractable log-expectation formulation. Motivated by the connection between guided diffusion sampling and constrained RL, the authors propose a more principled approach to estimating this intermediate energy term, which prior work has typically approximated or manually designed. Specifically, they linearize the Q-function via a first-order Taylor expansion and apply the moment-generating function of the Gaussian distribution to obtain an explicit and tractable estimation of the intermediate energy during diffusion sampling. This method, termed Analytic Energy-guided Policy Optimization (AEPO), bridges the gap between theory and practice. Empirically, AEPO demonstrates SoTA performance on most tasks in the D4RL benchmark, while maintaining computational efficiency comparable to existing approaches.

All reviewers have recognized the technical contribution to be important. The rebuttal further strengthened the submission by (1) clarifying the validity of the Taylor approximation through the smoothness of the SiLU activation, which ensures bounded errors, and (2) demonstrating robustness to the rescaled guidance parameter. These clarifications have addressed serveral reviewer's critical concerns regarding both the theoretical foundation and the practical impact of the method. Overall, the paper makes a principled, technically solid, and well-validated contribution.